# Diurnal variability of stratospheric column NO₂ measured using direct solar and lunar spectra over Table Mountain, California (34.38°N)

King-Fai Li[1], Ryan Khoury[1], Thomas J. Pongetti[2], Stanley P. Sander[2], Franklin P. Mills[3], Yuk L. Yung[2,4]

[1]Department of Environmental Science, University of California, Riverside, California, USA
[2]Jet Propulsion Laboratory, California Institute of Technology, Pasadena, California, USA
[3]Fenner School of Environment and Society, Australian National University, Canberra, Australia Capital Territory, Australia
[4]Division of Geological and Planetary Sciences, California Institute of Technology, Pasadena, California, USA

*Correspondence to*: King-Fai Li (king-fai.li@ucr.edu)

**Abstract.** A full diurnal measurement of stratospheric column NO₂ has been made over the Jet Propulsion Laboratory's Table Mountain Facility (TMF) located in the mountains above Los Angeles, California, USA (2.286 km above mean sea level, 34.38°N, 117.68°W). During a representative week in October 2018, a grating spectrometer measured the telluric NO₂ absorptions in direct solar and lunar spectra. The stratospheric column NO₂ is retrieved using a modified minimum-amount Langley extrapolation, which enables us to accurately treat the non-constant NO₂ diurnal cycle abundance and the effects of tropospheric pollution near the measurement site. The measured 24-hour cycle of stratospheric column NO₂ on clean days agrees with a 1-D photochemical model calculation, including the monotonic changes during daytime and nighttime due to the exchange with the N₂O₅ reservoir and the abrupt changes at sunrise and sunset due to the activation or deactivation of the NO₂ photodissociation. The observed daytime NO₂ increasing rate is $(1.34 \pm 0.24) \times 10^{14}$ cm$^{-2}$ h$^{-1}$. The observed NO₂ in one of the afternoons during the measurement period was much higher than the model simulation, implying the influence of urban pollution from nearby counties. A 24-hour back-trajectory analysis shows that the wind first came from inland in the northeast and reached the southern Los Angeles before it turned northeast and finally arrived TMF, allowing it to pick up pollutants from Riverside County, Orange County, and Downtown Los Angeles.

## 1 Introduction

Nitrogen dioxide (NO₂) plays a dominant role in the ozone (O₃)-destroying catalytic cycle (Crutzen, 1970). NO₂ column abundance has been measured using ground-based instruments since the mid-1970s [Network for the Detection of Atmospheric Composition Change (NDACC), http://www.ndacc.org] (e.g., Hofmann et al., 1995; Piters et al., 2012; Roscoe et al., 1999; Roscoe et al., 2010; Vandaele et al., 2005; Kreher et al., 2020), which serve as the standards for validating satellite measurements. Noxon (1975) and Noxon et al. (1979) retrieved the stratospheric NO₂ column by differential optical absorption spectroscopy (DOAS) in the visible spectral range using ratios of scattered sunlight from the sky and direct sun/moonlight at low (noon/midnight) and high (twilight) air mass factors over Fritz Peak, Colorado (39.9°N). Since the optical path of

sun/moonlight at dawn or dusk (solar/lunar zenith angle ≈ 90°) is much longer than the optical path of the direct sunlight at noon/midnight, the $NO_2$ absorption in the noon/midnight spectrum can be assumed to be small and the $NO_2$ absorption in the twilight slant column could therefore be isolated effectively by ratioing the scattered twilight spectrum to the scattered noon spectrum. This DOAS principle also applies to ratios of direct moonlight or sunlight at low and high air mass factors. Noxon

et al.'s (1979) measurements revealed sharp changes of the stratospheric $NO_2$ column before and after sunsets and sunrises at mid-latitudes. Similar DOAS measurements at high latitudes in the 1980s focused on the role of $NO_x$ in controlling $O_3$ and active halogen species in the polar stratosphere (Fiedler et al., 1993; Flaud et al., 1988; Keys and Johnston, 1986; Solomon, 1999). Johnston and McKenzie (1989) and Johnston et al. (1992) reported a reduction in the southern hemispheric $NO_2$ over Lauder, New Zealand (45.0°S), following the eruptions of El Chichón (in 1982) and Pinatubo (in 1991), respectively.

NO$_2$ column abundance has also been measured using direct solar spectra acquired by Fourier-Transform infrared (FTIR) spectrometers. Advantages of direct solar measurements are the lack of Raman scattering in the spectra, air mass factors determined geometrically rather than through a radiative transfer code, and provision of $NO_2$ column abundances at most times during the day. Sussmann et al. (2005) retrieved the stratospheric $NO_2$ column abundance over Zugspitze, Germany (47°N) using the infrared absorption in the solar spectrum near 3.43 μm. The stratospheric $NO_2$ column abundance was then

subtracted from the total column estimated from satellite measurements to obtain the tropospheric column. Wang et al. (2010) demonstrated how high spectral resolution measurements using a Fourier transform spectrometer could perform absolute $NO_2$ column abundance retrievals without the need for a solar reference spectrum. Because of the solar rotation, the Fraunhofer features in the UV spectra acquired simultaneously from the east and west limbs of the solar disk are Doppler shifted while the telluric $NO_2$ absorptions are not shifted (Iwagami et al., 1995). Thus, the telluric $NO_2$ absorptions can be identified by

correcting the Doppler shift without the need of an *a priori* solar spectrum. Other techniques, such as balloon-based *in situ* measurements (May and Webster, 1990; Moreau et al., 2005), balloon-based solar occultations (Camy-Peyret, 1995), as well as ground-based Differential Optical Absorption Spectroscopy: MAX-DOAS (Hönninger et al., 2004; Sanders et al., 1993), Direct Sun DOAS (Herman et al., 2009; Spinei et al., 2014; Herman et al., 2019) that have been actively applied in NDACC and the Pandonia Global Network. The DOAS techniques have also been employed to further characterize the vertical

distributions of $NO_2$ (Kreher et al., 2020).

Here we retrieve the stratospheric column $NO_2$ over Table Mountain Facility (TMF) in Wrightwood, California, USA (2.286 km above mean sea level, 34.38°N, 117.68°W) using Langley extrapolation to determine the reference spectrum and considering both daytime and nighttime chemistry. Daytime $NO_2$ concentration remains significant, albeit small relative to the night-time concentration, and varies from morning to afternoon. This daytime variation has been a source of error in

determination of the DOAS reference spectrum using Langley extrapolation. Comprehensive assessment of $NO_2$ must include both daytime and nighttime values. We therefore also retrieve daytime column $NO_2$ by acquiring direct sun spectra throughout the day. We will compare the daytime and nighttime stratospheric column $NO_2$ with those simulated in a one-dimensional (1-D) photochemical model. The effect of urban pollution on the measured $NO_2$ can be deduced from this comparison.

## 2 Data and Method

### 2.1 Instrumentation and measurement technique

The grating spectrometer used for the $NO_2$ spectral measurement is similar to the one used by Chen et al. (2011) and is installed in the same observatory. A heliostat and a telescope are used to direct and launch light into a fibre optic bundle placed at the focal plane of the telescope (Figure 1). The bundle consists of 19 silica fibres, 200 μm in diameter, arranged in a circular configuration (in SMA 905 connectors) on the source end and in a linear pattern on the spectrograph end. Before entering the spectrograph, light is passed through an order sorting filter (Schott GG-400 glass) and a shutter. The imaging spectrograph is a Princeton Instruments SP-2-300i with a 0.3-m focal length used with a 1200 g mm$^{-1}$ grating blazed at 500 nm. A CCD detector (Princeton Instruments PIXIS 400B) is placed at the focal plane of the spectrograph. The 1340 × 400 imaging array of 20 × 20 μm$^2$ pixels are vacuum sealed and thermoelectrically cooled to −80 °C.

Our assessment showed that 2 days away from the full moon would decrease the measured lunar intensity by ~20%. Therefore, in the following analysis, we only focus on acquired direct moon and direct sun spectra that are within 5 to 7 days of the full moon in order to maximize the signal-to-noise ratio (SNR). In addition, to minimize the terminator effects near sunrise/sunset, we use measurements with lunar/solar zenith angles less than 80°. The stray light is typically of the order of $10^{-4}$–$10^{-3}$.

When direct sunlight is measured, two ground glass diffuser plates are inserted into the beam prior to the telescope primary to integrate over the entire solar disk and to attenuate light. Additional attenuation of light to avoid detector saturation is accomplished by placing a 23% open area screen in the beam just after the diffuser plates. Overall, the solar throughput is reduced by a factor of ~$1.3×10^{-5}$. The resulting spectrum has a spectral grid spacing on the detector of 0.048 nm from 411 nm to 475 nm with a measured line shape of 0.34-nm FWHM sampled at ~7 pixels. Spectral calibration and line shape measurements are accomplished using a diffuse reflection of an Argon lamp near the fibre end, which gives a nearly linear result between pixel and wavelength with a small second order correction; the second order correction is considered in the calibration and the QDOAS fitting (see next section).

When direct moonlight is measured, the diffuser plates are removed. Since the sun is ~400,000 times the intensity of the full moon, the ratio between the light hitting our detector for solar noon (after inserting the diffuser plates) and lunar noon during the full moon is ~5. To maintain an approximately constant solar and lunar signal-to-noise ratio and fitting residuals, we vary the exposure time during specific times of solar and lunar noon, typically around ~3 s for lunar noon and ~0.6 s for solar noon, giving a ratio of ~5 to homogenize the solar and lunar photon counts mentioned above. At higher zenith angles, longer exposures were taken to keep the detector counts in the same range. The data were dark-corrected and averaged to obtain the desired signal levels; for the sun, this was consistently ~4 minutes; for the moon, the averaging time varied from ~8 minutes during the night of the full moon to 24 minutes on the night 3 days from full moon.

We estimate the SNR by assuming that the standard deviation of the difference of two consecutive spectra is close to the noise and that the average intensity of the two consecutive spectra is the signal. As a result, the SNR at full moon and solar

transits are ~2900 and ~4900, respectively. During the low sun/moon observations the SNR is more difficult to measure directly. However, the fitting residuals are consistent with these estimates.

**2.2 The DOAS retrieval**

The DOAS technique is used to retrieve the $NO_2$ slant column (Noxon, 1975; Noxon et al., 1979; Platt et al., 1979; Stutz and Platt, 1996). A spectrum measured by the grating spectrometer at any time of the day is ratioed to a pre-selected reference spectrum. From the ratioed spectrum, we retrieve the differential slant column $NO_2$ relative to the column that is represented by the reference spectrum. The total slant column is then the sum of the differential slant column and the reference column.

Our reference spectrum is a solar spectrum measured at the TMF ground level at local noon (Chen et al., 2011). This solar reference spectrum is used to ratio all other spectra collected, including those during the solar and lunar measurement cycles. In principle, one can retrieve the reference $NO_2$ column from the reference spectrum. However, this requires precise knowledge of the solar spectrum at the top of the atmosphere in order to isolate the $NO_2$ absorption. We will use a variant of the Langley extrapolation to circumvent the need of the retrieval of the reference column (Lee et al., 1994; Herman et al.,

2009); see following section for details.

The differential slant column $NO_2$ is retrieved by fitting the ratioed spectrum in a smaller window between 430 and 468 nm. This window has stronger $NO_2$ absorptions relative to other wavelengths in the instrument range (411–475 nm); see Figure 4 of Spinei et al. (2014). In addition, this window also has less interfering absorption from species other than the $O_3$, $O_4$ ($O_2$ dimer), and $H_2O$ (see below).

The spectral fitting is accomplished through the Marquardt-Levenberg minimization using QDOAS 3.2 (released in September 2017) retrieval software (http://uv-vis.aeronomie.be/software/QDOAS/). The high-resolution $NO_2$ absorption cross-sections at $T = 215$ K, 229 K, 249 K, 273 K, 298 K, and 299 K based on Nizkorodov et al. (2004) are convolved to the instrument resolution using the instrument line shape function and the Voigt line shape prior to its use in QDOAS. The yearly average from the TMF temperature LIDAR measurements are used to derive a reference for each altitude level by linear

interpolation between each adjacent cross-section, which is also adjusted for pressure broadening using the results of Nizkorodov et al. (2004). We use 3rd order polynomials for broadband and offset. Some studies, such as Herman et al. (2009), use 4th or higher order polynomials for wider spectral windows. Since the $NO_2$ absorption features are much narrower than our spectral window (430–468 nm), the broad shape of the 3rd order polynomial does not affect the $NO_2$ retrievals. In addition, for our spectral window, we tested our retrieval algorithm using a linear baseline and we concluded that a 3rd order polynomial

reduces the residuals more effectively than a linear baseline. All five cross-sections were used to create a single $NO_2$ reference. Our $NO_2$ cross section reference assumes the yearly average temperature profile at TMF and a low level of free tropospheric $NO_2$. The effective temperation of the $NO_2$ absorption cross section used in the work is 231 K. To test the sensitivity of these assumptions we considered two extreme cases: (i) a cooler atmosphere with a lower partition of $NO_2$ in the free troposphere and (ii) a warmer atmosphere with a higher partition of $NO_2$ in the free troposphere. The effective temperatures of these two

cases are estimated by 229 K and 249 K, respectively.  The difference between retrievals using these extreme cases is ~5%; the regular variation of temperature and tropospheric $NO_2$ at TMF is well within estimates. Each level's reference is then multiplied by a weight which is proportional to the standard atmosphere and then summed to obtain a single reference used in the fitting.   In addition to $NO_2$, other absorptions by $O_3$, $O_4$ ($O_2$ dimer), and $H_2O$ in the same spectral window are simultaneously retrieved. The $O_3$ cross section is from Serdyuchenko et al. (2014) for 11 temperature references ranging from

193 K to 293 K. Like $NO_2$, all 11 cross-sections are used in the spectral fitting for $O_3$. In contrast, for $O_4$ and $H_2O$, only a single temperature reference is used. The $O_4$ cross-sections are from Thalman and Volkamer (2013) at 273 K. The $H_2O$ cross-sections at 296 K are from HITRAN 2016 (Gordon et al., 2017). Figure 2 shows an example of a fitted spectrum on October 24, 2018. The $NO_2$ abundance retrieved from QDOAS is the desired differential slant column $NO_2$ relative to our chosen reference spectrum.

140       The 2-$\sigma$ uncertainty due to the spectral fitting residual lies between $0.1\times10^{15}$ molecules $cm^{-2}$ and $0.6\times10^{15}$ molecules $cm^{-2}$, with a mean of ~$0.4\times10^{15}$ molecules $cm^{-2}$, which is equivalent to a mean of 10% uncertainty. The distribution of the retrieval uncertainty is shown in Figure 3 (inset).

      The air mass factor is calculated using secant of the solar/lunar zenith angle. Herman et al. (2009) considered an altitude correction of the air mass factor. The altitude correction is generally negligible except for zenith angles $\geq 80°$ but we

do not make measurements at those zenith angles (see §2.1).

## 2.3 The modified minimum-amount Langley extrapolation (MMLE)

      Let $y$ be the differential slant column $NO_2$ along the line-of-sight, $y_0$ the reference column $NO_2$, $m$ the stratospheric airmass factor (which is proportional to the geometric secant of the solar zenith angle in the stratosphere for these direct solar and lunar observations), and $x$ the total vertical column $NO_2$; $x$ is our target quantity. The differential slant column can be

approximated as the total vertical column multiplied by the stratospheric airmass factor after the subtraction of the reference column:

$$y = m\,x - y_0 \tag{1}$$

If $y_0$ were known, then $x$ would be simply $m^{-1}(y + y_0)$. The Langley extrapolation technique for determination of the extra-terrestrial reference obtains $x$ and $-y_0$ as the slope and the intercept of the linear regression of $y$ against $m$, respectively, assuming $x$ is temporally constant (i.e. the vertical column does not change during the course of the day). In this formulism, the reference column $y_0$ is an extrapolated value corresponding to hypothetical zero airmass ($m = 0$).

      The Langley extrapolation was first used to measure the solar spectrum at the top of the atmosphere (Langley, 1903)

and has also been used to measure atmospheric constituents (e.g., Jeong et al., 2018; Toledano et al., 2018; Barreto et al., 2017; Huber et al., 1995; Bhartia et al., 1995). However, the assumption of a constant $x$ is often violated due to diurnal variabilities

in the atmospheric constituents driven by, *e.g.*, the incident solar radiation, transmittance, dynamics, and human activities. In our case, the afternoon stratospheric column $NO_2$ is greater than the morning stratospheric column $NO_2$ (see our Figure 4). Several modifications have been proposed to relax the assumption of a constant $x$ (e.g., Ångström, 1970; Shaw, 1976; Long and Ackerman, 2000; Cachorro et al., 2008; Kreuter et al., 2013; Marenco, 2007). In this work, we combine the modifications used in Lee et al. (1994) and Herman et al. (2009) to account for the effects due to the stratospheric $NO_2$ diurnal variability and urban pollution.

Lee et al. (1994) replaced the constant $x$ with an *a priori* function of $m$, denoted by $x_a(m)$:

$$y = \alpha \, m \, x_a(m) - y_0, \tag{2}$$

Eq. (2) is analogous to Eq. (1) except that now $y$ is regressed against the product $m \, x_a(m)$. $\alpha$ is the slope of the regression line and it serves as an effective scaling factor that adjusts the chemical rates in the *a priori* knowledge. Eq. (2) presents a modified Langley extrapolation. The *y*-intercept, $y_0$, obtained from the modified Langley extrapolation is then used to derived the observed total vertical column through the transformation $m^{-1}(y + y_0)$. Note that $\alpha$ is not used in this transformation.

As in Lee et al. (1994), assuming the chemical processes of $NO_2$ are much faster than the dynamical processes so that the $NO_2$ diurnal cycle is at photochemical equilibrium, we obtain $x_a(m)$ from a 1-D photochemical model (to be described in the next section). The $x_a(m)$ we use corresponds to a clean atmosphere only. To perform the regression, we plot $y$ against the product $m \, x_a(m)$ (Figure 3, blue open circles). If all $NO_2$ columns are measured on clean days, then they would ideally fall on a straight line (which, apart from the natural variability in the background, holds true for Lee et al.'s (1994) measurements over Antarctica). However, if there is a pollution source near a measurement site, like the TMF, then some of the measured $NO_2$ column may be significantly higher than $x_a(m)$, leading to a large vertical spread in the scattered plot. The pollution-induced deviation from $x_a(m)$ may be highly variable, depending on the source types and the meteorology. When a large number of measured $NO_2$ columns on clean and polluted days are plotted together against $m \, x_a(m)$, the baseline of the scattered data may be considered as the background $NO_2$ diurnal cycle in a clean atmosphere (Herman et al., 2009). Herman et al. (2009) called their method the minimim-amount Langley extrapolation (MLE). Expanding on their terminology, we call our method, which combines the MLE with Lee et al.s' modification, the modified MLE, or MMLE. Note, however, that the MMLE differs from the optimal estimation that is commonly used in satellite retrieval, where the statistics of priori knowledge is used to constrain the retrieved value; no prior constraint is used in the MMLE.

Our measurements made during October (a non-summer season) were mostly under unpolluted conditions (see §3.5). Thus, we applied the MMLE to derive a baseline for an estimation of the background $NO_2$ diurnal cycle, which is then used in the regression with the modelled diurnal cycle. On the Langley plot (Figure 3), we divide the range of $m \, x_a(m)$ (from $4.5 \times 10^{15}$ to $3 \times 10^{16}$ molecules cm$^{-2}$ during our campaign) into 20 equal bins. To be consistent with the 2-$\sigma$ spectral-fit uncertainty, we use the 10-percentile of the $y$ distribtion in each bin to define a baseline (Figure 3, green dots).

Note that the data points are sparsely distributed at high air mass factors in Figure 3. This is because while the measurements were made at relatively uniform time intervals, the air mass factor $m = sec\theta$ effectively stretch the time intervals at high air mass factors. The number of data points in the bins drops progressively by a factor of ~2: the counts drop exponentially from 431 in the first bin, $(4.5–6)\times10^{15}$ molecules cm$^{-2}$, to only 12 in the bin $(1.5–1.65)\times10^{16}$ molecules cm$^{-2}$. The determination of the 10-percentile for bins with centers greater than $1.5\times10^{16}$ molecules cm$^{-2}$ is then subject to large

uncertainties. Since mathematically, the 10-percentiles at high air mass factors (i.e. at the edge of the data distribution) have higher effects on a linear fit, the resultant Langley extrapolation would be strongly biased by the uncertainties of the 10-percentiles at high mass factors. Thus, to obtain a linear fit for the Langley extrapolation, we apply more weights to bins with more data counts. This definition of the weights should mimic the reduction of the variance of a sample mean by the factor of $\frac{1}{N}$ (or $\frac{1}{\sqrt{N}}$ for the standard deviation of a sample mean). Thererfore, we define the weight as unity for the first bin, $(4.5–6)\times10^{15}$

molecules cm$^{-2}$. The weight for the second bin, $(6–7.5)\times10^{15}$ molecules cm$^{-2}$, is the ratio of the data counts of this bin over the first bin. The weight for the third bin is the ratio of the data counts of this bin over the second bin, etc. The weighted linear fit obtained using these weights is used for the Langley extrapolation. Figure 3 compares the Langley extrapolations using the weighted (solid red line) and unweighted linear fit (dashed red line). Since the 10-percentiles at high air mass ($\geq 1.5\times10^{16}$ molecules cm$^{-2}$) are generally overestimated due to insufficient data counts, the unweighted linear fit tends to have a steeper

slope, leading to a ~15% higher reference column ($5.44\times10^{15}$ molecules cm$^{-2}$) relative to the weighted linear fit. This overestimation of the reference column may create an artifact in the diurnal cycle due to the normalization factor $m^{-1}(y + y_0)$.

    The weighted Langley extrapolation (Figure 3, solid red line) provides the values of $\alpha$ and $y_0$ for our stratospheric column NO$_2$ estimation. The weighted fit gives $\alpha = 0.80 \pm 0.04$ and $y_0 = (4.74 \pm 0.21) \times 10^{15}$ molecules cm$^{-2}$ (at 2-$\sigma$ levels). This value of $y_0$ is our reference column used for both daytime and nighttime measurements. We estimate the total

retrieval uncertainty to be the root-mean-square of the spectral fitting uncertainty and the uncertainty in $y_0$, which is ~$0.5\times10^{15}$ molecules cm$^{-2}$ (2-$\sigma$).

### 2.4 The photochemical model

    Our $x_a(m)$ is based on the Caltech/JPL 1-D photochemical model (Allen et al., 1984; Allen et al., 1981; Wang et al., 2020), shown as the black solid line in Figure 4. This photochemical model includes the stratospheric species that are important

for O$_3$, odd-nitrogen (NO$_x$ = N + NO + NO$_2$ + NO$_3$ + 2N$_2$O$_5$) and odd-hydrogen (HO$_x$ = H + OH + HO$_2$) chemistry, including the reactions discussed in §3.1. Nitrous oxide (N$_2$O) is the main parent molecule of NO$_2$ in the lower stratosphere. The concentration of N$_2$O at the ground level of the model is fixed at 330 ppb (https://www.esrl.noaa.gov/gmd/hats/combined/N2O.html). The kinetic rate constants are obtained from the 2019 JPL Evaluation (Burkholder et al., 2019).

The sunrise/sunset times and the solar noontime in the model are calculated using the ephemeris time. We use Newcomb parameterizations of the perturbations due to the Sun, Mercury, Venus, Mars, Jupiter, and Saturn (Newcomb, 1898).

We also use Woolard parameterizations for the nutation angle and rate (Woolard, 1953). More modern calculation of the ephemeris time may be used (e.g., Folkner et al., 2014) but the difference in the resulting ephemeris time is small (less than 0.1 s) and does not significantly impact our model simulation.

We progress the model in time until the diurnal cycle of the stratospheric $NO_2$ becomes stationary. Throughout the progression, the pressure and temperature profiles are fixed and do not vary with time. The model latitude is set at 34.38°N and the model day is set as October 26. The column $NO_2$ is the vertical integral of the $NO_2$ concentration. The simulation represents the stratospheric $NO_2$ abundance in a clean atmosphere without tropospheric sources.

## 3 Results and Discussions

### 3.1 Diurnal variation in stratospheric column $NO_2$

Figure 4 presents our preliminary observational data (colour dots) obtained during October 23–28, 2018. During the measurements, the skies were mostly clear or only partly cloudy, so we were able to make continuous solar spectral measurements throughout the whole period. During October, the local sunrise and sunset time were around 07:00 PST and 18:00 PST, respectively. At sunrise and sunset, the ambient twilight in the background of the moonlight occultation should be

accounted for in the $NO_2$ retrieval, which is beyond the scope of this work. For this work, we exclude lunar $NO_2$ data when the ambient scattered twilight, including those from civil sources, is significant, which typically occurs when the lunar elevation angle is less than 6° above the horizon. Figure 4a shows the daily diurnal cycles during the week of measurements and Figure 4b shows the aggregated diurnal cycle as a function of local time. The solid black line in both panels is the simulated 24-hour cycle of the stratospheric column $NO_2$ variability in the 1-D model. The dashed line in Figure 4b is a second simulation

with a slightly lower temperature (see §3.4). Overall, the baseline simulation captures the observed trends during the daytime and the nighttime. The observations reveal day-to-day variability, but our back-trajectory analysis shows that the day-to-day variations during October 23–26 and 28 are likely due to natural variability of the background in the north while that on October 27 is likely due to urban sources from the Los Angeles basin in the south (see §3.5)

On most days, the stratospheric column $NO_2$ over TMF increased from $\sim 2 \times 10^{15}$ molecules cm$^{-2}$ in the morning to

$\sim 3.5 \times 10^{15}$ molecules cm$^{-2}$ in the evening. There are 3 main sources of $NO_x$ contributing to the daytime increase. The ultimate source is the reaction of $N_2O$ with excited oxygen $O(^1D)$ resulting from the photolysis of $O_3$ in the stratosphere between 20–60 km, which produces nitric oxide (NO) molecules and eventually $NO_2$ through the $NO_x$ cycle aided by $O_3$:

$$N_2O + O(^1D) \rightarrow NO + NO, \qquad (R1)$$

$$NO + O_3 \rightarrow NO_2 + O_2. \qquad (R2)$$

Another major source is the photolysis of the reservoir species, nitric acid ($HNO_3$) and dinitrogen pentoxide ($N_2O_5$):

$$HNO_3 + h\nu \rightarrow NO_2 + OH, \tag{R3}$$

$$N_2O_5 + h\nu \rightarrow NO_2 + NO_3. \tag{R4}$$

There is also a small source due to the photolysis of $NO_3$:

$$NO_3 + h\nu \rightarrow NO_2 + O, \tag{R5}$$

but this source is not significant due to the low $NO_3$ abundance during daytime. $NO_2$ is converted back into NO through the reaction with oxygen atom (O) in the upper stratosphere (above 40 km):

$$NO_2 + O \rightarrow NO + O_2 \tag{R6}$$

or via photolysis below 40 km:

$$NO_2 + h\nu \rightarrow NO + O. \tag{R7}$$

But since NO and $NO_2$ are quickly interconverted within the $NO_x$ family, Reactions R6 and R7 do not contribute to a net loss of $NO_2$. The ultimate daytime loss of $NO_2$ is the reaction with the hydroxyl radicals (OH) that forms $HNO_3$, which may be transported to the troposphere, followed by rainout:

$$NO_2 + OH + M \rightarrow HNO_3 + M. \tag{R8}$$

The significant deviation of daytime $NO_2$ from the model simulation on October 27 was likely due to urban pollution (see §3.5).

At sunset, the photolytic destruction (Reaction R7) in the upper stratosphere terminates while the conversion of NO (Reaction R2) continues in the lower stratosphere. Meanwhile, the production of O is significantly reduced, which also reduces the loss of $NO_2$ via Reaction R6. As a result, the stratospheric column $NO_2$ increases by a factor of ~3 at sunset.

Next, the stratospheric column $NO_2$ decreases from $\sim 6.5 \times 10^{15}$ molecules cm$^{-2}$ after sunset to $\sim 4.5 \times 10^{15}$ molecules cm$^{-2}$ before sunrise. During nighttime, $NO_2$ is converted to $N_2O_5$ via the reaction with $O_3$ and $NO_3$:

$$NO_2 + O_3 \rightarrow NO_3 + O_2, \tag{R9}$$

$$NO_2 + NO_3 + M \rightarrow N_2O_5 + M. \tag{R10}$$

Most $N_2O_5$ stays throughout the night, although there is a small portion that thermally dissociates back to $NO_2$ and $NO_3$. Thus, the net effect is a secular decrease in nighttime $NO_2$.

Finally, at sunrise, photolytic reactions resume, resulting in an abrupt decrease in the total $NO_2$ column by a factor of ~2 due to Reactions R6 and R7.

### 3.2 Vertical profile of $NO_2$ production and loss

To better understand the contributing factors of the variability of stratospheric column $NO_2$, we show the simulated vertical $NO_2$ profile in Figure 5. The $NO_2$ concentration is dominant between 20 km and 40 km (Figure 5a). At noontime, the model $NO_2$ profile has a peak of $\sim 1.7 \times 10^9$ molecules $cm^{-3}$ at 30 km (Figure 5a, orange line). At mid-night, the $NO_2$ concentration is much higher throughout the stratosphere. The corresponding peak has a larger value of $\sim 2.4 \times 10^9$ molecules $cm^{-3}$ and is shifted slightly upward to 32 km (Figure 5a, green line). Therefore, the stratospheric column $NO_2$ is dominated by the variability near 30 km.

The diurnal cycles of the $NO_2$ concentration at altitudes between 14 km and 38 km are shown in Figure 5b. These cycles show that the daytime increase and the nighttime decrease occur only in the lower stratosphere between 18 km and 34 km. At other altitudes, the daytime and nighttime $NO_2$ concentrations are relatively constant. The $NO_2$ cycles closely resemble those of $N_2O_5$. Figure 6 shows the $N_2O_5$ concentrations between 14 km and 34 km. During daytime, $N_2O_5$ is photolyzed into $NO_2$ and $NO_3$ through Reaction R4, leading to an increase in the daytime $NO_2$; during nighttime, $NO_2$ is thermally converted into $N_2O_5$ through Reactions R9 and R10, leading to a decrease in the nighttime $NO_2$. Figure 6 shows that the conversion between the reservoir and $NO_2$ dominates between 18 km and 34 km, consistent with the $NO_2$ diurnal cycles. In particular, the quadratic decreasing trend of the daytime $N_2O_5$ is consistent with the quadratic increasing trend of the daytime $NO_2$. Therefore, the secular $NO_2$ changes during daytime and nighttime are dominated by $N_2O_5$ conversions.

### 3.3 Daytime $NO_2$ increasing rate

Reactions (R1)–(R5) contribute the daytime increase of $NO_2$. Sussmann et al. (2005) first obtained a daytime $NO_2$ increasing rate from ground-based measurements. They reported an annually averaged value of $(1.02 \pm 0.06) \times 10^{14}$ $cm^{-2}$ $h^{-1}$ over Zugspitze, Germany (2.96 km, 47°N). For October alone, they obtained a value of $(1.20 \pm 0.57) \times 10^{14}$ $cm^{-2}$ $h^{-1}$. For comparison, we calculate the daytime increasing rate using our data between 7 AM and 4 PM. To obtain a rate corresponding to a clean atmosphere, we define a baseline of the diurnal cycle using the 10-percentile in the 30-minute bins from 7 AM to 4 PM (Figure 7). This results in a total of 19 bins, which is ~half of the number of points in October shown in Figure 4a of Sussmann et al. (2005). We then apply the linear regression to the baseline and obtain an increasing rate of $(1.34 \pm 0.24) \times 10^{14}$ $cm^{-2}$ $h^{-1}$ in October over TMF (34.4°N). Thus our value is consistent with Sussmann et al.'s (2005) value.

**3.4 Temperature sensitivity**

While the 1-D model simulation captures most of the observed diurnal variability, the rate of decrease in $NO_2$ during nighttime is slightly overestimated in the model. Here we explore a possible uncertainty due to the prescribed temperature profile.

The chemical kinetic rates in the model are dependent on temperature. The temperature profile that has been used to obtain the baseline diurnal cycle corresponds to a zonal mean temperature profile at the equinox and 30° latitude (Figure 8, solid line). To test the sensitivity of the simulated 24-hour cycle of $NO_2$, we reduce the input temperature below 60 km by 5 K (Figure 8, dashed line). Note that the 5 K reduction is much larger than the observed tidal variation in stratospheric temperature below 50° latitude, which is ~0.1 K in the lower stratosphere and ~1 K in the middle stratosphere (Sakazaki et al., 2012). We choose this exaggerated reduction in order to clearly show the temperature effect on the $NO_2$ chemistry.

Figure 4b (dash-dotted line) shows the simulated stratospheric $NO_2$ column using the reduced temperature profile. Because of the reduction in temperature, the nighttime loss due to the reactions with $O_3$ and $NO_3$ through Reactions R9 and R10 is slower. As a result, the simulated nighttime $NO_2$ is higher than the baseline simulation but the rate of decrease agrees better with the observations. On the other hand, due to the less efficient reaction $NO + O_3$, the simulated daytime $NO_2$ is slightly lower than the baseline simulation but it still agrees with the daytime observation. Thus, while the equinox temperature profile used in the baseline run is sufficient for the simulation of the $NO_2$ diurnal cycle, we do not exclude possible effects of temperature uncertainties on the nighttime simulation.

**3.5 Back-trajectories**

Since the TMF is located at the top of a mountain in a remote area, high values of $NO_2$ measured on October 27, 2018, were likely due to atmospheric transport of urban pollutants from nearby cities, especially the Los Angeles megacity. While chemical processes would quantitatively alter the amount of $NO_2$ to be observed over TMF, a back-trajectory study suffices to provide evidence on how the urban pollutants may be transported to TMF.

Figure 9 shows the 24-hour back-trajectories that eventually reached TMF (2.286 km above sea level) at 3 PM during the observational period. These back-trajectories are calculated using the National Oceanic and Atmospheric Administration (NOAA)'s Hybrid Single Particle Lagrangian Integrated Trajectory (HYSPLIT) model (Stein et al., 2015). We use wind fields from the National Centres for Environmental Prediction (NCEP)'s North American Mesoscale (NAM) assimilation at a horizontal resolution of 12 km. To illustrate the wind speed, we plot the 6-hour intervals using the black dots on the trajectories.

The trajectories on 4 of the 6 days (October 23–26) during the observational period converged towards TMF from inland in the north and the east. These inland areas are behind the San Gabriel and San Bernardino Mountain Ranges and are shielded from the urbanized Los Angeles basin. Therefore, the stratospheric column $NO_2$ measured over TMF on these days closely follow the clean atmosphere simulated by the 1-D model. The trajectories on the other 2 days (October 27–28) converged towards TMF from the Los Angeles basin in the southwest. But these 2 trajectories were very different. The back-

trajectory of October 27 (Figure 9, orange) started going southwestward from the Mojave Desert north of the San Bernardino Mountains at the 24-hour point and passed across the Riverside Basin between the Santa Ana Mountains and San Jacinto Mountains at 18-hour point. The Riverside Basin is one of the most polluted areas in the United States. Then the trajectory continued southwest to pass across the Orange County at the 12-hour point before it turned northwestward towards Downtown Los Angeles at the 6-hour point. Finally, the trajectory turned northeastward and reached TMF. The wind speed over the Los Angeles basin on October 27 was slower than those in other days, favouring more accumulation of pollutants over the Basin. Thus, the 24-hour back-trajectory on October 27 transported the pollutants in the Riverside Basin and the Los Angeles basin, resulting a significant surplus of $NO_2$ in the TMF observation as seen in Figure 4. In contrast, the trajectory on October 28, (Figure 9, purple) came directly from the Pacific Ocean at a relatively high speed, spending only ~4 hours in the Los Angeles basin before reaching TMF. However, our measurement on October 28 stopped at noon due to a change in instruments and we are unable to verify whether the urban source would elevate the total column $NO_2$ in that evening.

## 4 Summary

We have presented the diurnal measurements of stratospheric column $NO_2$ that has been made over the TMF located in Wrightwood, California (2.286 km, 34.38°N, 117.68°W) from October 23 to October 28, 2018. The instrument measures the differential slant column $NO_2$ relative a reference spectrum at the noontime. To retrieve stratospheric column $NO_2$ in the reference spectrum, we applied a variant of the Langley extrapolation. The conventional Langley extrapolation assumes a constant column throughout the day, which does not hold for $NO_2$. To properly consider the time-dependence of $NO_2$, we combine two methods independently developed by Lee et al. (1994) and Herman et al. (2009). The combined method, called the modified minimum-amount Langley extrapolation (MMLE), first obtains a baseline of the observed diurnal cycle, which is assumed to be the diurnal cycle in a clean atmosphere. Then the baseline is fitted against the modelled diurnal cycle in a 1-D photochemical model so that the stratospheric column $NO_2$ in the reference spectrum is given by the *y*-intercept of the fitted line.

The measured 24-hour cycle of the TMF stratospheric column $NO_2$ on clean days agrees well with a 1-D photochemical model calculation. Our model simulation suggests that the observed monotonic increase of daytime $NO_2$ is primarily due to the photodissociation of $N_2O_5$ in the reservoir. From our measurements, we obtained a daytime $NO_2$ increasing rate of $(1.31 \pm 0.41) \times 10^{14}$ $cm^{-2}$ $h^{-1}$, which is consistent with the value observed by Sussmann et al. (2005), who reported a daytime $NO_2$ increasing rate of $(1.20 \pm 0.57) \times 10^{14}$ over Zugspitze, Germany (2.96 km, 47°N). Our model also suggests that during nighttime, the monotonic decrease of $NO_2$ is primarily due to the production of $N_2O_5$. Furthermore, the abrupt $NO_2$ decrease and increase at sunrise and subset, respectively, are due to the activation and deactivation of the $NO_2$ photodissociation.

The observed $NO_2$ in the afternoon on October 27, 2018 was much higher than the model simulation. We conducted a 24-hour HYSPLIT back-trajectory analysis to study how urban pollutants were transported from the Los Angeles basin. The

back-trajectories in 4 of the 6 days during the measurement period went directly from inland desert areas to the TMF. The back-trajectory in another day came from the southwest coastline, spending less than 6 hours over the Los Angeles basin before reaching the TMF. Lastly, the 24-hour back-trajectory on October 27, 2018 was characterized by a unique slow wind that came from inland in the northeast and spent more than 18 hours in the Los Angeles basin, picking up pollutants from Riverside, Orange County, and finally Downtown Los Angeles before reaching TMF.

**Appendix A. Comparison of the modified MLE (MMLE) with the standard MLE**

      The MMLE is used to account for the diurnal asymmetry of the stratospheric $NO_2$ column before the Langley extrapolation is applied. To illustrate the necessity of the removal of the diurnal asymmetry, consider a single day of observed stratospheric column $NO_2$. Figure A1a plots the observations on October 25, 2018 against the air mass factor ($AMF = \sec\theta$) as in a standard MLE. Based on our back-trajectory analysis, the atmosphere above TMF on October 25, 2018 should have

little urban $NO_2$ contamination. Both solar (pale orange dots) and lunar (pale blue dots) data exhibit U-shapes that is due to the secular increase and decrease during the daytime and the nighttime, respectively. For the solar data, the AM data lies on the lower arm of the U-shape and the PM data lies on the upper arm. For the lunar data, the reverse is true: data before sunrise lie on the upper arm of the U shape and data after sunset lie on the lower arm.

      To perform a Langley extrapolation for the data shown in Figure A1a, one needs to decide which of the four arms to

be used for the linear regression model $y = a\,AMF + b$. The Principle of Minimum-amount suggests that we should start with the lowest arm, i.e. the daytime AM data. Note that in order to obtain the straight line passing through the 10-percentile baseline, we have ignored the points before noon (around 10 AM to 11:30 AM), i.e. points located around the bottom of the U-shape. If we use the observations between 6 AM and 10 AM, we obtain the purple line in Figure A1a, which gives a $y$-intercept of $(-4.12 \pm 0.14) \times 10^{15}$ molecules cm$^{-2}$.

405       The above Langley extrapolation, however, does not take any of the daytime PM and all lunar data into account. In particular, the daytime PM data should also be used to define a minimum-amount profile, given the fact that the atmosphere was mostly clean on that day. Suppose we perform another Langley extrapolation using the daytime PM data between 12 PM and 5 PM (rose line). The resultant $y$-intercept is $(-5.25 \pm 0.27) \times 10^{15}$ molecules cm$^{-2}$ (2-$\sigma$), which is statistically different from the value obtained using the daytime AM data. A reasonable estimate of the $y$-intercept is then the average of the two

values, which is $(-4.69 \pm 0.21) \times 10^{15}$ molecules cm$^{-2}$.

      Finally, since the wind on the TMF is mostly downhill during autumn, the lunar data also correspond to a clean atmosphere and should also be used to derive the $y$-intercept. If we use all four arms in Figure A1a, then the average value of the $y$-intercept is $(-4.36 \pm 0.25) \times 10^{15}$ molecules cm$^{-2}$, where the uncertainty is the root-mean-squares of the uncertainties of the four values.

415       In the above calculation, the ignorance of the data points near the bottom of the "U"-shape has excluded a large number of observations near local solar/lunar noon and thus the resultant $y$-intercept is biased by high zenith angles. It is not

clear how the data near the solar/lunar noon may be kept in the standard MLE due to the assumption of the linearity in AMF. As a result, a zenith angle-dependent Langley extrapolation model needs to be developed.

The above example shows that the determination of the *y*-intercept of the standard MLE is not straightforward when
(i) the background $NO_2$ has secular trends in daytime and nighttime and (ii) the daytime and nighttime abundances are different before and after the terminator. In contrast, the MMLE approach we have developed in this work minimizes the background diurnal asymmetry, so that the "regularized" data points almost form a straight line (Figure A1b) when they are plotted against the modelled diurnal cycle. The linear regression model $y = a\,m\,x_a + b$, where $m\,x_a$ is the modelled slant column $NO_2$, can be applied to all data points, regardless of the time of the day or whether the data point is a solar or lunar measurement. With
this modified MLE, the regressed *y*-intercept is $(-5.22 \pm 0.14) \times 10^{15}$ molecules cm$^{-2}$, which is statistically different from the average of the values derived from the four arms in the standard MLE approach.

The issue with the standard MLE is exacerbated when observations on multiple days are plotted against the AMF. The U-shape may be smeared vertically into a continuum (Figure A1c). The smearing, in our case, are primarily due to natural variability of the background, except for October 27 when the observed $NO_2$ appears above the continuum of the daytime data
due to the urban pollution. As a result, while we are still able to define the minimum-amount profile (10-percentile) for the daytime AM data, the determination of the minimum-amount profiles of the daytime PM and the lunar data are difficult. This leaves us the daytime AM data alone for the Langley extrapolation (red line) but, as shown above, the resultant *y*-intercept $[(-4.10 \pm 0.46) \times 10^{15}$ molecules cm$^{-2}]$ may be biased.

In contrast, the observed data points still almost form a straight line in the MMLE approach when they are plotted
against the modelled diurnal cycle (Figure A1d). This allows the determination of the minimum-amount profile using all solar and lunar measurements (raspberry line). With the weighted Langley extrapolation described in §2.3, the resultant *y*-intercept, $(-4.74 \pm 0.21) \times 10^{15}$ molecules cm$^{-2}$, is again statistically different from the one obtained using the standard MLE approach.

### Appendix B. Effects of spherical geometry in the 1D model

The diurnal cycle simulated in the 1D model (Figure 3) is calculated assuming a plane-parallel atmosphere, where
the times of the sunrise and the sunset do not depend on altitude. For a more realistic simulation, we have conducted another calculation using the spherical geometry, so that the terminator chemistry is dependent on altitude. Figure B1 compare the simulated diurnal cycles in a plane-parallel atmosphere and in a spherical atmosphere. The difference between the two diurnal cycles is the largest in the evening but it is much smaller than the spread of the observations due to the natural variability. Therefore, the simulation with a plane-parallel atmosphere is adequate to provide a theoretical diurnal cycle for the modified
Langley extrapolation.

**Acknowledgements.** The assistance of George Mount (Washington State University) in Langley analysis is greatly appreciated. KFL thanks Sally Newman and Tracy Xia (Bay Area Air Quality Management District) for their assistance in

setting up and executing the HYSPLIT model. RK was an undergraduate research assistant under the supervision of KFL. We thank Ralf Sussmann for handling our manuscript and providing useful comments.

**Financial support.** Support from the NASA SAGE-III/ISS Validation, Upper Atmosphere Research and Tropospheric Composition Programs is gratefully acknowledged. YLY was supported in part by NASA grant P1847132 via UCLA. RK thanks the generous supports from the AGU Student Travel Grant Award and the Marsh Environmental Sciences Travel Award by the University of California, Riverside.

**Data availability.** The differential slant column $NO_2$ used in this paper can be obtained from the supplement of this article.

**Supplement.** The supplement related to this article is available online.

**Author contributions.** KFL and TJP prepared the manuscript, with significant conceptual input from SPS and YLY, and critical feedback from all the co-authors. SPS and TJP designed and operated the instrument at the Table Mountain. TJP retrieved the slant column $NO_2$ from the spectra and developed the Langley method. KFL and YLY performed the model simulations. RK analysed some of the observational and model data.

**Competing interests.** The authors declare that they have no conflict of interest.

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

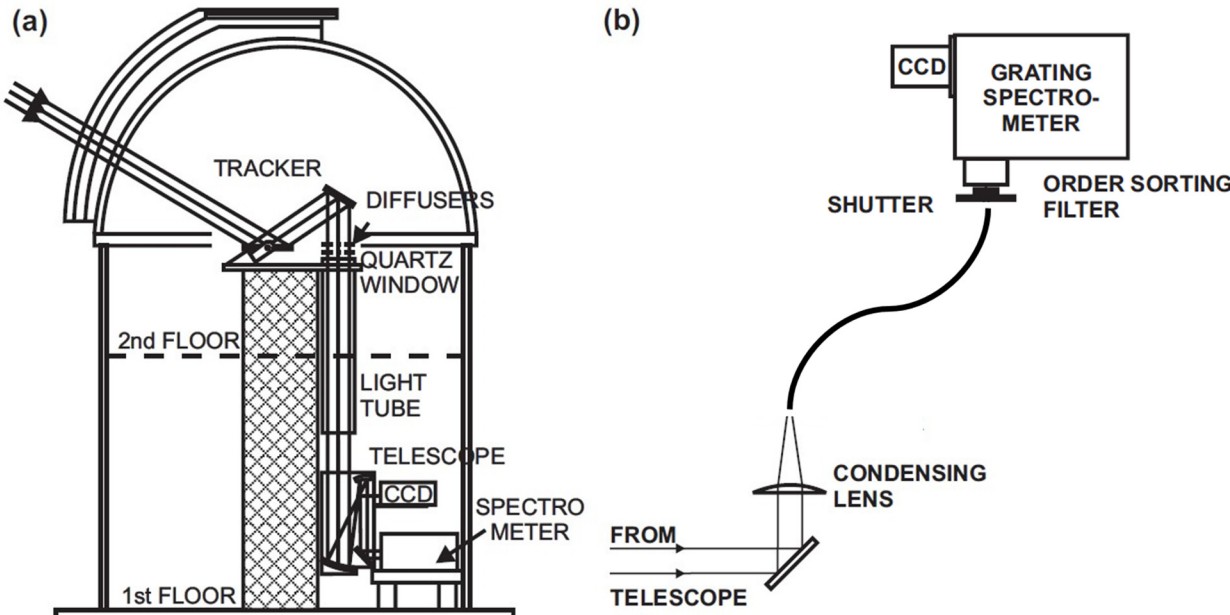

**Figure 1: Schematic of the instrument light path over the Table Mountain Facilities (TMF, 2.286 km above mean sea level, 34.38°N, 117.68°W), Wrightwood, California, USA.** (a) Light is collected by the primary of the heliostat (tracker), reflected down to the telescope on the first floor which conditions it to a 7-cm diameter beam. (b) The light is then reflected to a condensing lens into a fibre optic bundle, past a shutter, order-sorting filter, and then into the spectrometer. The fibre bundle contains 19 fibres in a round pattern at the entrance, and at the exit fibres are arranged in a line pattern that is set parallel to the spectrometer slit.


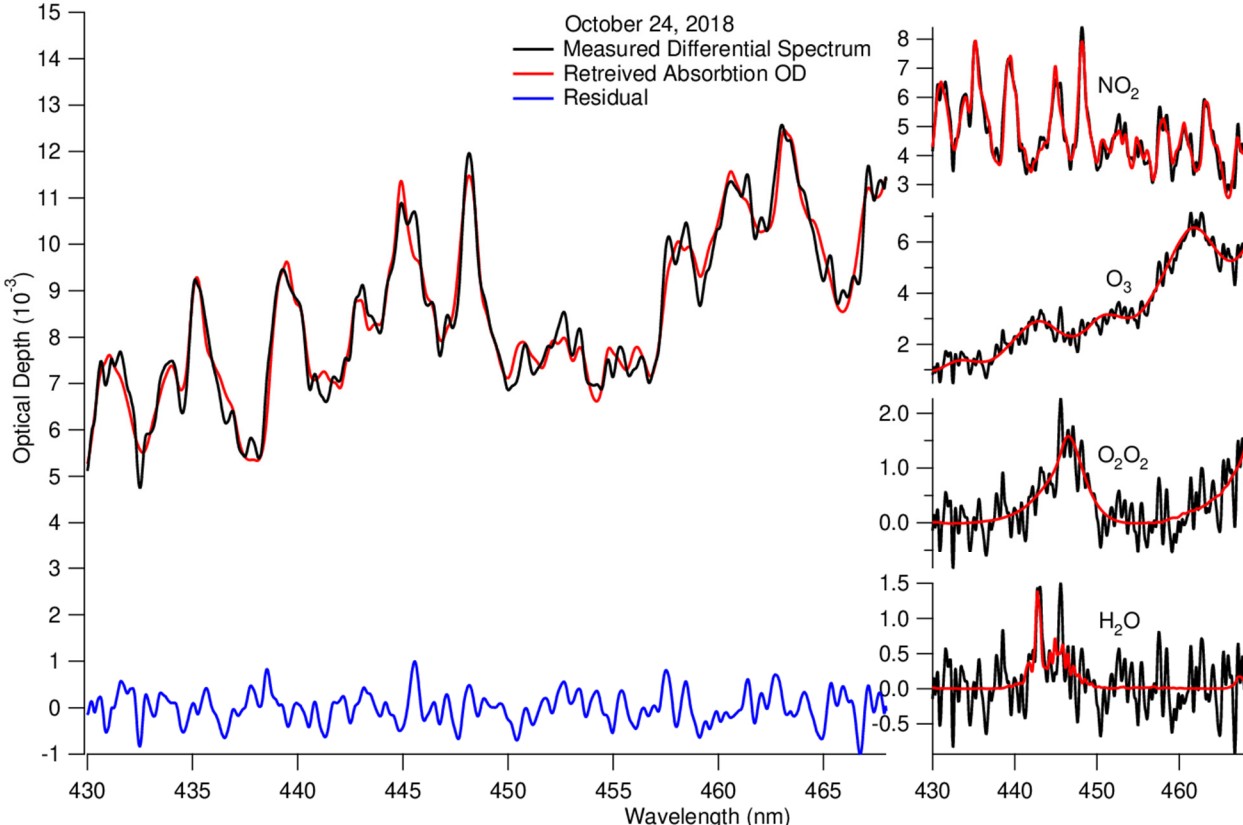


**Figure 2. A sample QDOAS spectral fit of a lunar spectrum at an air mass factor of 2.21 on October 24, 2018 at 7:25 PM.** The measured spectrum is shown by the black curve on the left panel. The fitted spectrum (red) is overlaid and the residual spectrum (blue) is shown at the bottom. Four species are considered in the spectral fit: $NO_2$, $O_3$, $O_4$, and $H_2O$. The spectral fits are performed simultaneously in QDOAS. The red lines on the right column are the fitted spectra of the corresponding species. To visualize the signal-to-noise ratios, we

add the residual spectrum (blue on the left panel) to individual fitted spectra, which are shown as the black spectra in the subpanels on the right.

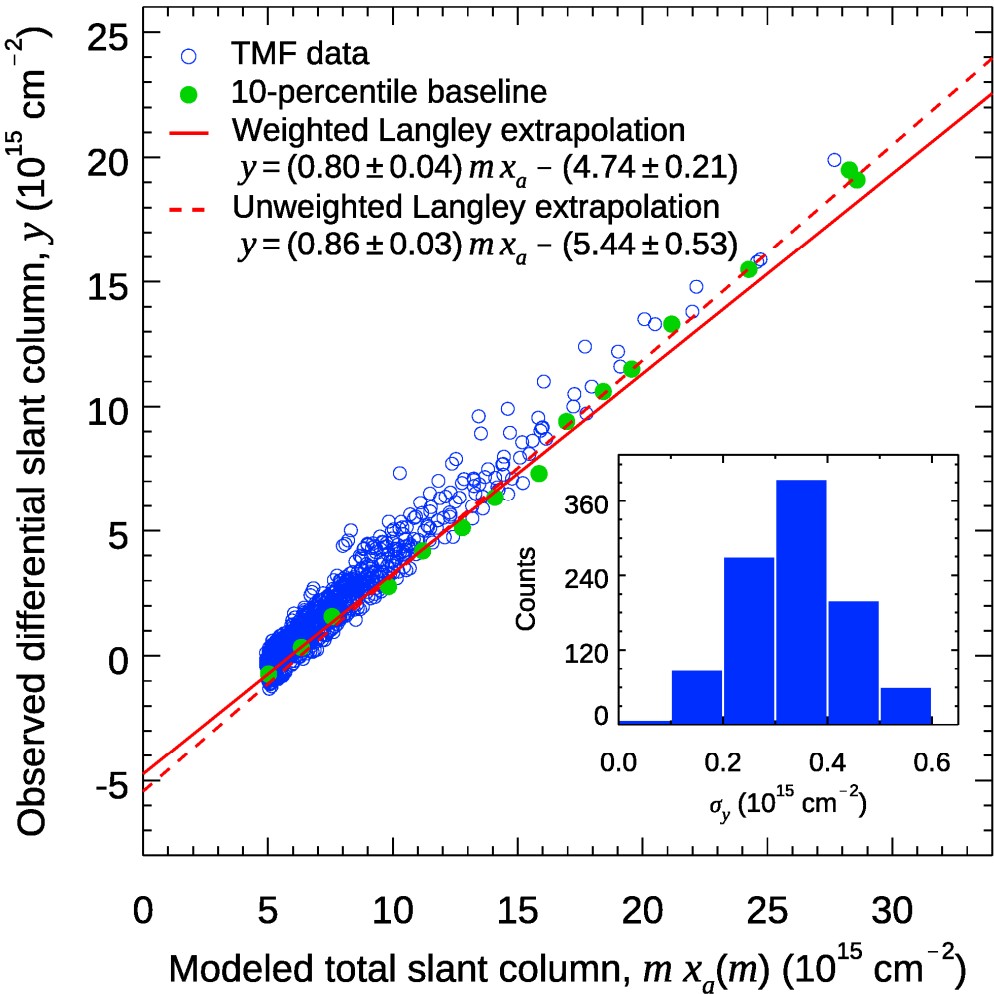

**Figure 3. The modified minimum-amount Langley extrapolation (MMLE).** The blue circles are the observed differential slant columns during our campaign over TMF from October 23 to October 28, 2018. Each observational value is plotted against the total slant column modelled at the same time of the day (e.g. 11:05 AM PST). The green dots are the 10-percentile of 20 uniform bins on the $x$-axis. The red line is a linear regression of the green dots, which is taken as the background diurnal cycle in a clean atmosphere. The linear fit weighted by the number of data points in the bins is $y = 0.80\,m\,x_a(m) - 4.74 \times 10^{15}$. The 2-$\sigma$ uncertainties of the slope and the intercept are **0.04** and $0.21 \times 10^{15}$, respectively. The $y$-intercept thus gives a reference column, $y_0 = 4.74 \times 10^{15}$ molecules cm$^{-2}$. The unweighted linear fit overestimates the reference column ($5.44 \times 10^{15}$ molecules cm$^{-2}$) because of the sparse data points at air mass factors greater than ~2. Thus, the weighted linear fit is used as the Langley extrapolation in this work. The inset shows the distribution of the 2-$\sigma$ uncertainty of the observed differential slant columns.

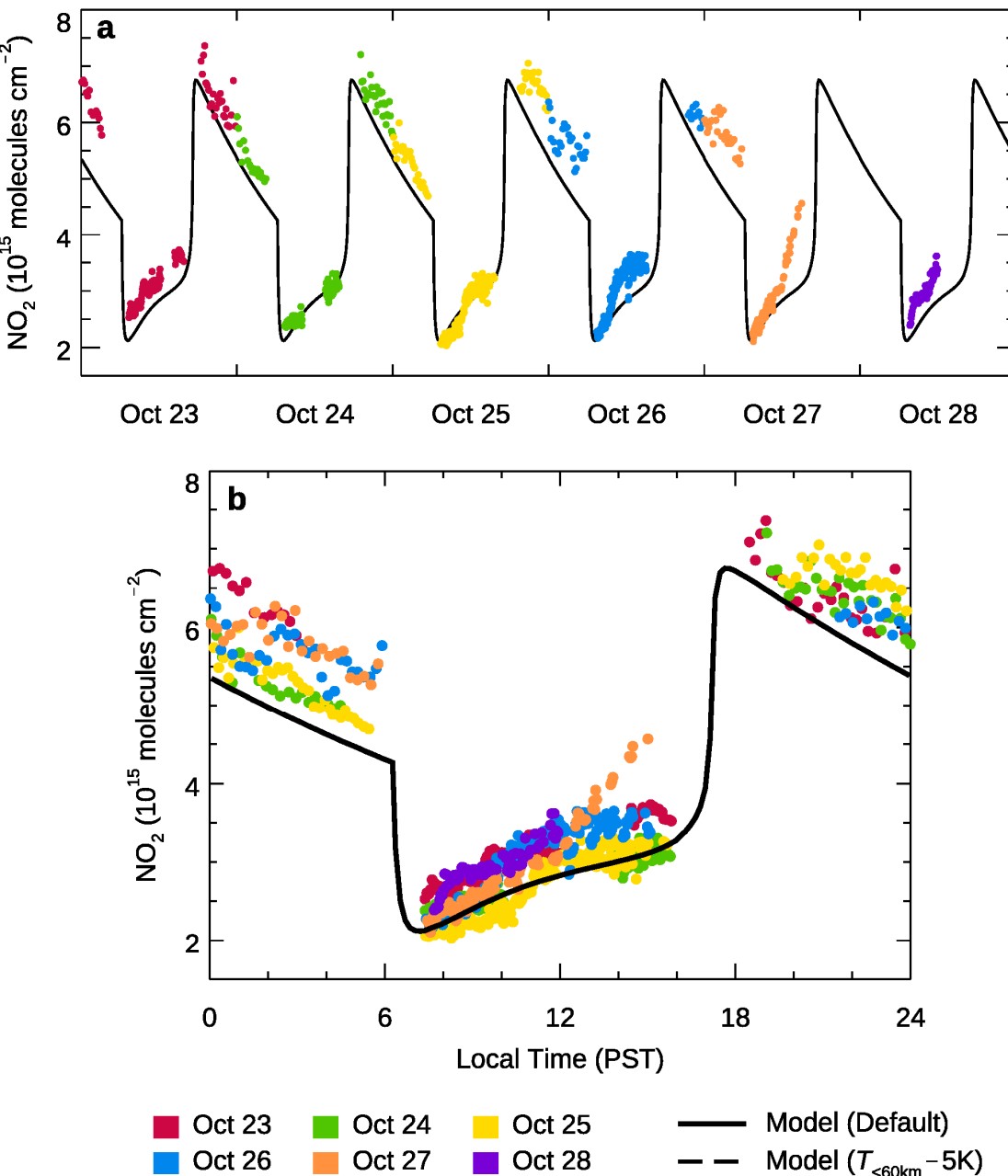

**Figure 4: The column NO₂ abundance measured over TMF on October 23−28, 2018, represented by the color dots.** The 1-D model simulation, with default input temperature and surface N₂O being 330 ppb, representing October 26 is shown as the solid black line. (a) The column NO₂ measurements on individual dates. (b) The aggregated column NO₂ measurements as a function of local time. An additional 1-D model simulation with temperature below 60 km reduced by 5 K, is shown as the dashed line.

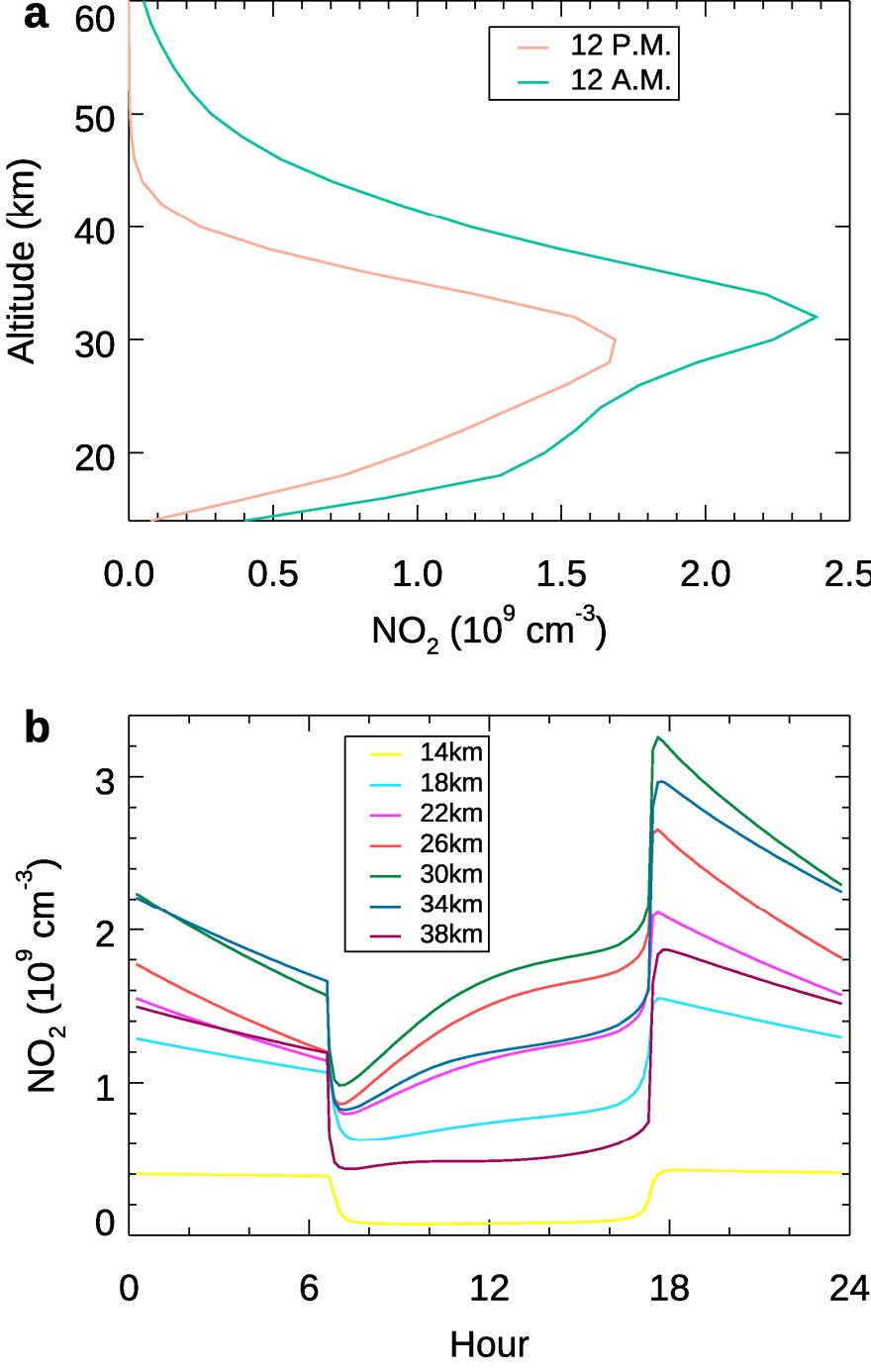


**Figure 5: Simulated vertical NO₂ concentration.** (a) The simulated NO$_2$ vertical concentration between 14−38 km at 00:00 PST (green) and 12:00 PST (orange) corresponding to October 27 in the 1-D photochemical model. (b) Same as (a) except the simulated NO$_2$ variation over the 24 hours at selected altitudes.

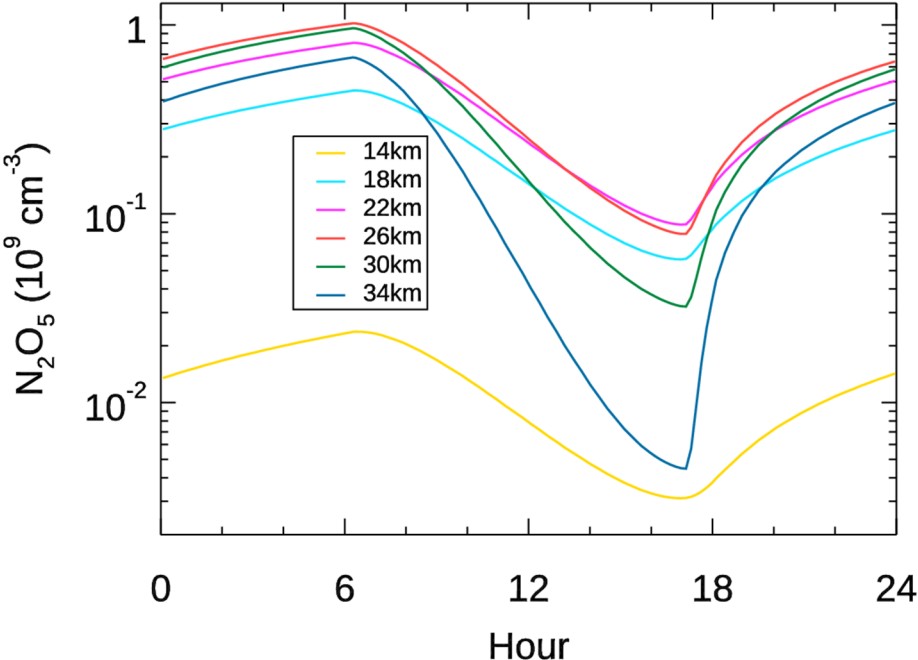

**Figure 6. Same as Figure 5(b) except for $N_2O_5$.**

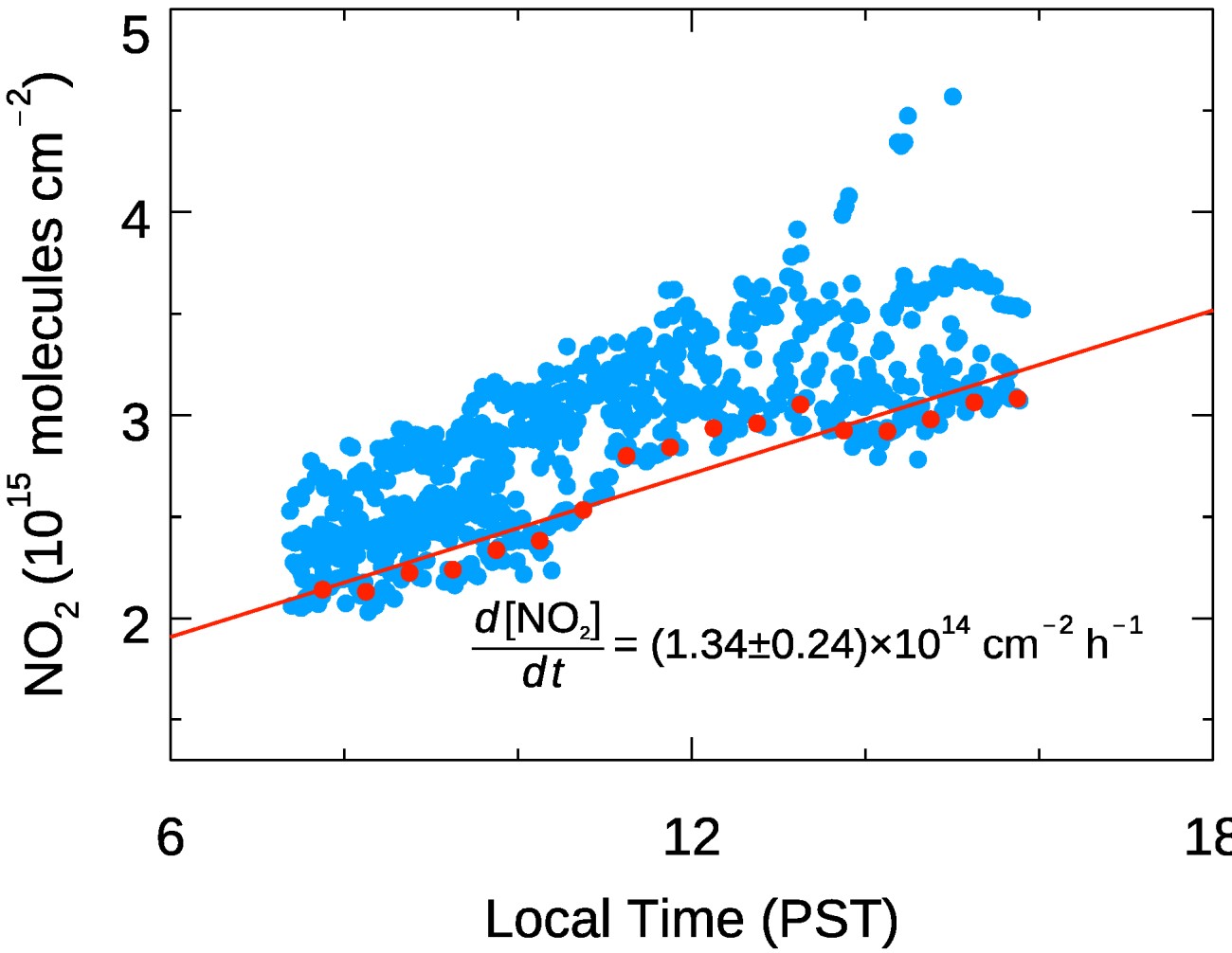

**Figure 7. The daytime NO₂ increase obtained from the baseline of the observed diurnal variability.** The blue points are the same as the daytime data shown in Figure 4. The red points are the 10-percentile of the daytime data in 30-minute intervals between 7 AM to 4 PM,
which form a baseline of the daytime variability. The daytime NO₂ increase rate, obtained from the linear regression of the red points, is $(\mathbf{1.34 \pm 0.24}) \times \mathbf{10^{14}}$ cm⁻² h⁻¹.

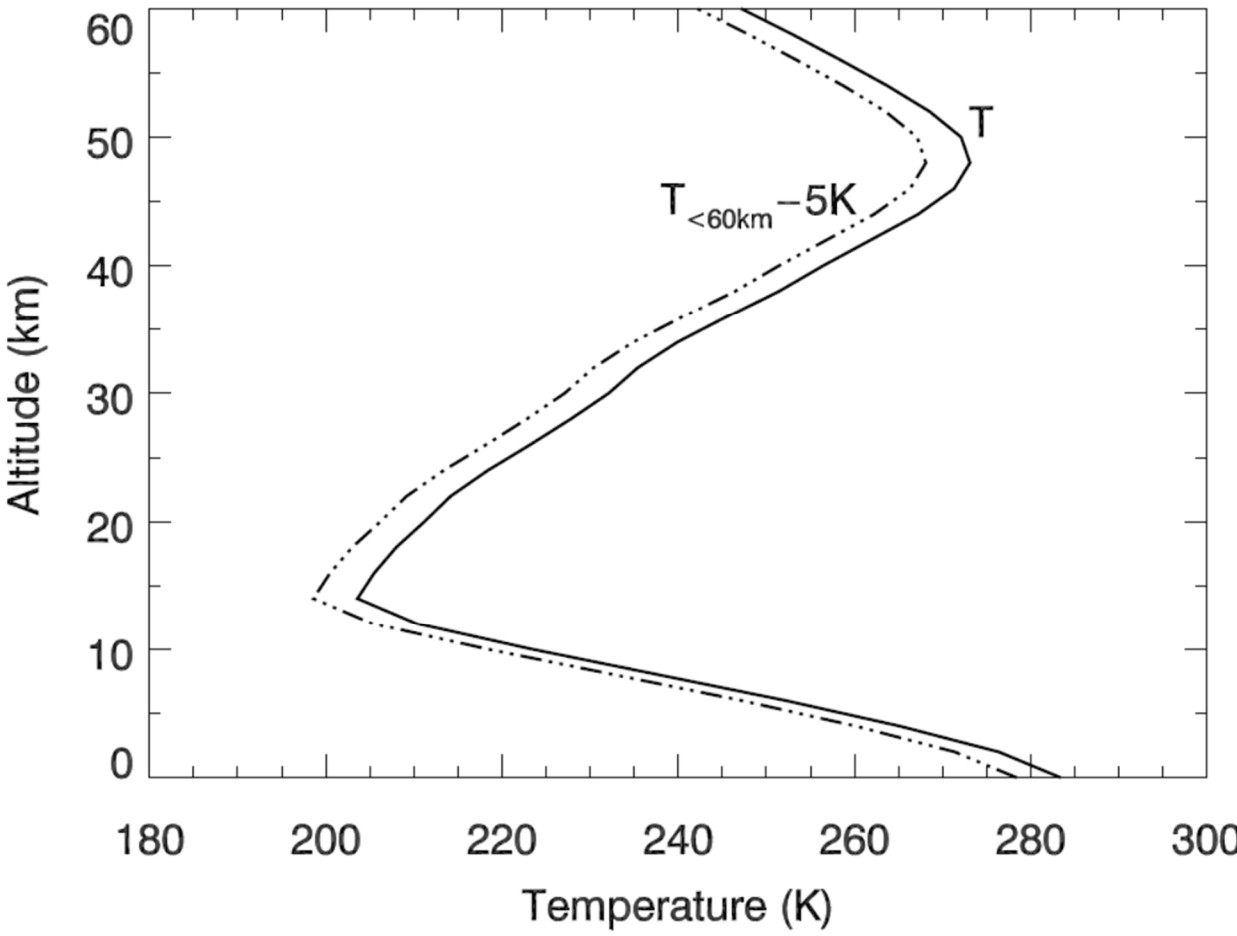

**Figure 8: The temperature profiles used in the 1-D Caltech/JPL photochemical model: the baseline profile (solid line) based on the equinox zonal average at 30° latitude and the modified profile where the temperature below 60 km is reduced by 5 K (dash-dotted line).**

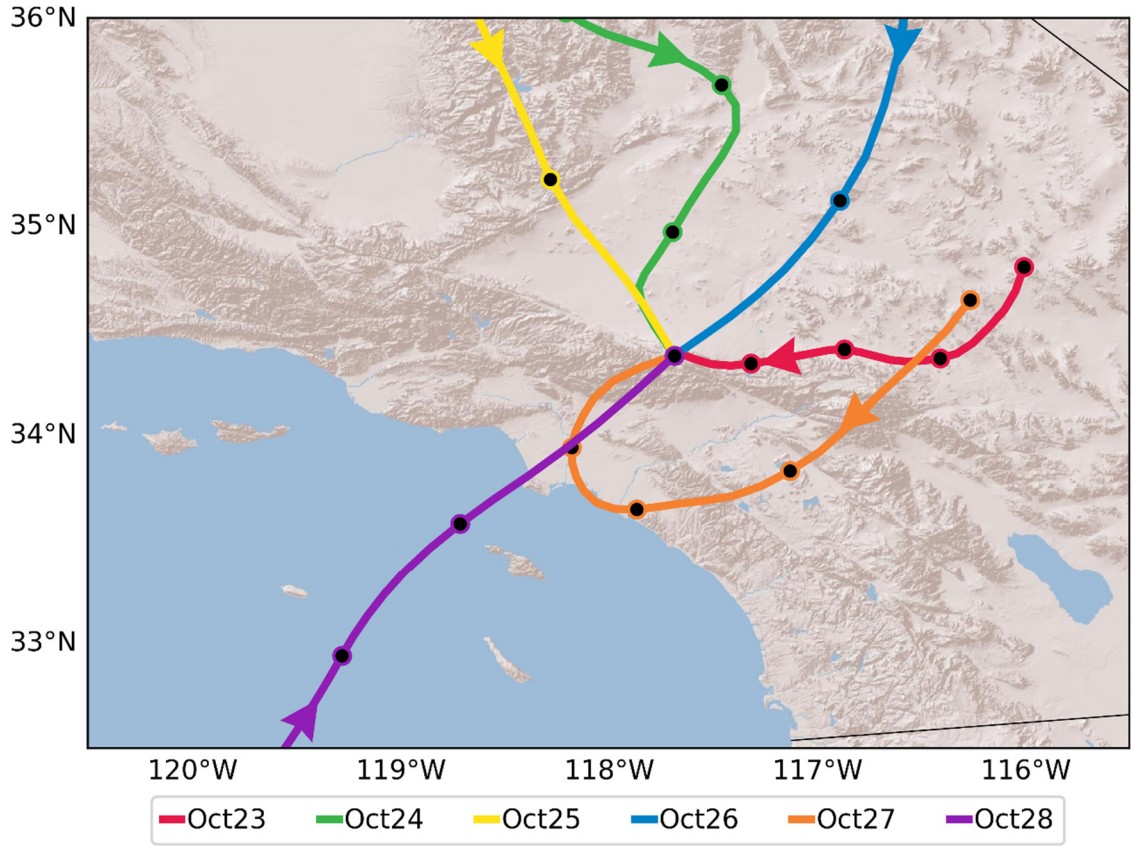

**Figure 9: The 24-hour back-trajectories of ambient air flow that reached TMF at 15:00 PST on each day from October 23 to October 28, 2018.** The colour codes are the same as those used in Figure 4. The black dots represent the 6-hour intervals on the trajectories.

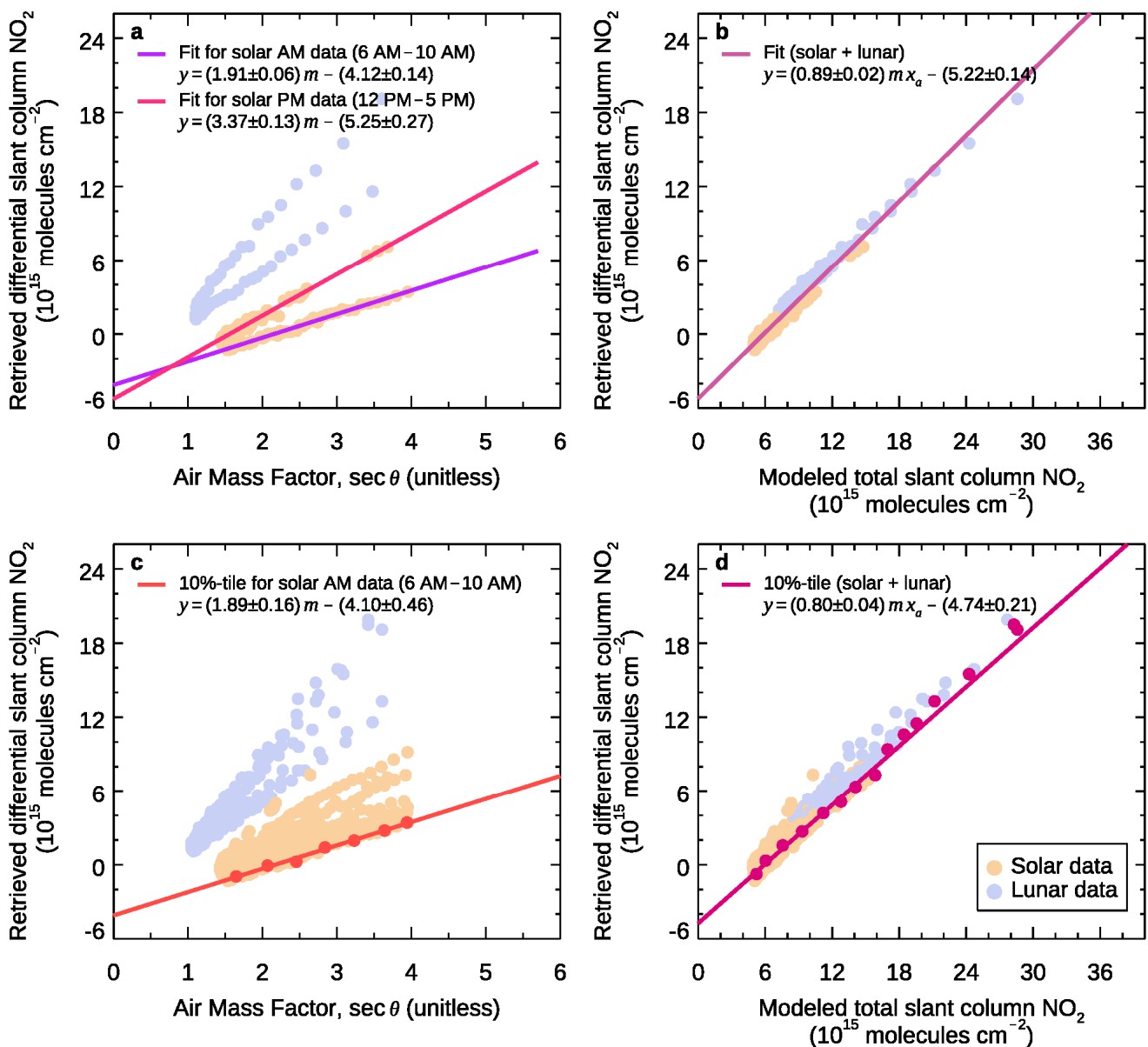

**Figure A1. Comparison of the standard MLE (a, c) and the modified MLE (MMLE) (b, d) introduced in this work for single-day (a, b) and multiple-day data (c, d).** Panel d is the same as Figure 3 except for the separation of the daytime and nighttime data.

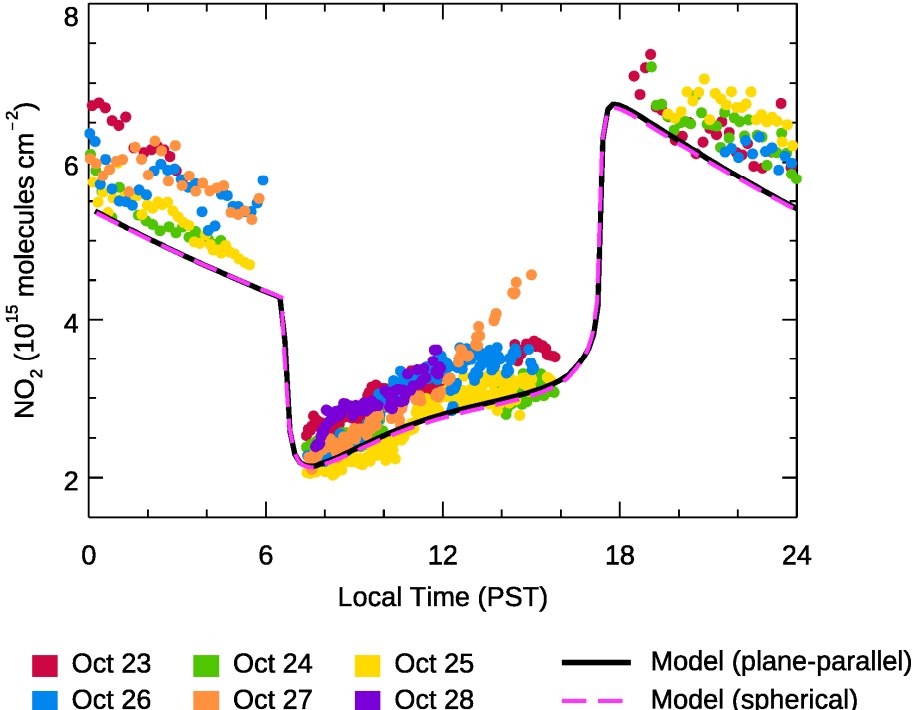

**Figure B1. Comparison of the simulated diurnal cycles of the stratospheric column NO₂ in a plane-parallel atmosphere and a spherical atmosphere.** The data points are the same as in Figure 4b.
