# Peer review of "Diurnal variability of stratospheric column NO2 measured using direct solar and lunar spectra over Table Mountain, California (34.38°N)"

_Atmospheric Measurement Techniques, 2020_

## Referee Comment (RC1) · Anonymous Referee #1 · 19 Jul 2020

This is a good paper, but the title is misleading even though it is accurate. For Table Mountain, a mostly NO2 free site, the diurnal variation is stratospheric, except for 1 day during the 1-week campaign. The title should say as much. "Stratospheric diurnal variability of NO2 measured and modelled using direct solar and lunar spectra over Table Mountain, California (34.38°N)".

Since the authors have 6 days of data, it would be interesting to have an extra figure showing the daily diurnal variation in addition to Figure 3. The agreement with the model run is quite good except after sunset (Fig. 3) when the magnitude and shape are different. Is there an explanation? I know this is not a modelling paper, but the

treatment of sunrise and sunset seems incomplete. There should be a time delay as a function of altitude with the sun reaching higher altitudes first. Since the authors are not showing data during sunrise and sunset, it does not matter much.

The linear fit in Figure 6 does not mean much, other than as a baseline, as there are two linear regimes, one from 07:00 to 13:00 and from 13:00 to 16:00 hours. Is there an explanation for the two regimes?

On the instrument: What is the stray light. What is the SNR of each measurement?

I know you are working with the standard QDOAS software, but could you give examples of the DOAS fitting and residuals.

The writing is clear with no significant errors. The figures are clear and easy to understand The title should be revised The paper should be published with only minor changes as described above

---

## Referee Comment (RC2) · Anonymous Referee #2 · 2 Sep 2020

King-Fai Li et al 2020 present direct sun and direct moon NO2 measurements over a high altitude mostly unpolluted site, JPL-TMF near Wrightwood, CA, during 6 days (around full moon) in October 2018. They proposed to combine two Langley-like techniques to estimate amount of NO2 atmospheric absorption during the reference spectrum measurement time. The proposed approach takes advantage of 1-D photochemical model to estimate diurnal variation in NO2 and minimum Langley extrapolation technique to reduce the effect of NO2 pollution. Modeling results are compared with the measurements. Chemical reactions for different processes are shown. Accurate measurements of diurnal NO2 variation are important and the topic fits into the scope of the "Atmospheric Measurement Techniques" journal.

[Figure]

Major comments:

Not sufficient measurements (only 6 days) were presented in the paper to apply minimum Langley extrapolation technique (MLE). MLE is a statistical method and requires sufficient data to "accouter" conditions with "constant" vertical columns at each solar zenith angle. This threshold was not met during this study. MLE uses as low percentile for fitting as possible (within SNR) to capture background NO2. Increasing percentile used for Langley fitting does not simply improves the statistics, as stated in the paper, but can significantly alter the result. This can be easily remediated by including more measurements, especially at this mostly clean site.

While the idea of improving estimation of slant column density in the reference spectrum using a 1-D photochemical model is appealing, the authors have not demonstrated that it provides a better result than MLE itself. Looking at the data in Fig. 2 and 3, MLE will most likely result in lower amount in the reference spectrum, and the final vertical columns will agree significantly better with the model diurnal change than the retrieved columns (but will have an offset). Authors need to show that the results are better than the MLE by itself, and for that more measurements are required. Note, that to determine amount in the reference spectrum, full moon is not needed, since the analysis is done on the direct sun data. It is not clear from the presented results if the error in the model simulations actually is smaller than the uncertainty in MLE. This needs to be demonstrated.

It is unclear what benefits this approach (1D stratospheric model) will have under the "persistent" pollution levels when the total NO2 abundance is dominated by anthropogenic emissions at all times.

Description of the DOAS fitting settings is not sufficient. What are polynomial orders (e.g. broad band, offset, wavelength shift), what sources of other gases cross sections and at what temperatures were used in the analysis? Were NO2 cross sections at all five temperatures used in the retrieval? If yes, how they were fitted and combined?

Why this fitting window was selected (430 and 468 nm)? How exactly air mass factor was calculated? What is the DOAS fitting quality of NO2 from direct sun and direct moon (residual OD)?

No error budget is presented for the measured NO2 columns.

The paper in general reads more like a modeling paper then the measurement paper.

There are routine NO2 stratospheric measurements conducted by the NDACC stations (zenith sky DOAS) and total column measurements of NO2 using direct sun and direct moon within Pandonia Global Network. They should be mentioned in the review of NO2 measurements. In general, citations tend to include mostly early works and not give current status.

It is unclear how the lunar measurement where taken. Lunar irradiance is about $10^6$ lower than solar irradiance. In this study, integration time for sun measurements is 16 times shorter then for moon. Difference in lunar vs solar measurements (diffusers, filters, etc), and what effect it has on spectrometer illumination should be presented. Target signal-to-noise ratio stated.

---

## Author Comment (AC1) · 30 Nov 2020

Dear Editor,

We would like to thank the two anonymous reviewers for their time and valuable comments that have greatly improved our manuscript, especially during the pandemic. All comments have been carefully considered and responded during our revision of the manuscript. Our responses have been submitted as a supplement in this Author Comment.

We look forward to hearing from you soon!

[Figure]

Best wishes,

King-Fai, with my co-authors

Please find our responses from the link below:

Please also note the supplement to this comment:
https://amt.copernicus.org/preprints/amt-2020-173/amt-2020-173-AC1-
supplement.pdf

**Supplement:**

**TABLE OF CONTENTS**

**Responses to Reviewer #1's comments**

Box 1.1

For Table Mountain, a mostly $NO_2$ free site, the diurnal variation is stratospheric, except for 1 day during the 1-week campaign. The title should say as much. "Stratospheric diurnal variability of $NO_2$ measured and modelled using direct solar and lunar spectra over Table Mountain, California (34.38°N)".

We thank the reviewer for this interesting comment. The retrieval technique used in this manuscript does return the total column above the Table Mountain (2.25 km), no matter how small the tropospheric contribution is. The phrase in the original title, "diurnal variability of total column $NO_2$", has an important purpose, which is to inform the reader that our retrieval is limited to the total column $NO_2$ only and that the diurnal variability is generally a sum of the tropospheric and stratospheric diurnal variability, regardless whether the atmosphere is clean or not. Given this limitation, a diagnostic tool, like the HYSPLIT model, needs to be employed to reveal the tropospheric contribution in some cases where high total column $NO_2$ is observed, like the one we have in the manuscript on Oct 27 (see new Figure 3). In contrast, if we use the phrase "stratospheric diurnal variability", we are afraid that the reader might have an impression that we had a retrieval technique that separate the stratospheric column from the total column.

Another suggestion made by the reviewer is to add in the title "and modelled" after "measured". We are afraid that the suggested title may be a little inaccurate because the diurnal variability is not modelled "using direct solar and lunar spectra". We thought of something like

> "Diurnal variability of total column $NO_2$ measured using direct solar and lunar spectra over Table Mountain, California (34.38°N) and modelled in a 1-D photochemical model"

but this title looks a bit long. We think that the original title contains the most important component of the research (retrieval based on direct solar/lunar spectra) and the reader will be able to learn about our modeling work from the abstract. We believe that the original title is a balance of multiple factors. We are happy to continue to consider further suggestions should the reviewer has other concerns about the original title.

Again, we greatly appreciate all suggestions made by the reviewer. We have considered these suggestions very seriously before we made our final decisions.

Box 1.2
Since the authors have 6 days of data, it would be interesting to have an extra figure showing the daily diurnal variation in addition to Figure 3.

We have added a new panel in Figure 3 showing the daily diurnal variations as suggested:

[Figure]
* * *
**Box 1.3**

The agreement with the model run is quite good except after sunset (Fig. 3) when the magnitude and shape are different. Is there an explanation?
* * *
We thank the reviewer for this insightful comment. Although we did not say explicitly in the manuscript, the temperature sensitivity test (Section 3.3) was intended to explore one possible cause of the difference between the model and the observation during the nighttime. The modelled rate of decrease at night with the default temperature is greater than observed (the solid line in Figure 3). Lowering the temperature by 5 K reduces the modelled rate of decrease (the dashed line). Thus, the temperature profile could be a possible reason for the different shapes at night. However, a more definitive study would be needed in future publications to investigate this problem. In response to this comment, we have added a paragraph at the beginning of Section 3.4:

> "While the 1-D model simulation captures most of the observed diurnal variability, the rate of decrease in the total $NO_2$ column during nighttime is slightly overestimated in the model. Here we explore a possible uncertainty due to the prescribed temperature profile."

and have revised the conclusion of the same section in Line 333 of the revised manuscript

> "Thus, while the equinox temperature profile used in the baseline run is sufficient for the simulation of the diurnal cycle of the $NO_2$ column, we do not exclude possible effects of temperature uncertainties on the nighttime simulation."
* * *
**Box 1.4**

I know this is not a modelling paper, but the treatment of sunrise and sunset seems incomplete. There should be a time delay as a function of altitude with the sun reaching higher altitudes first. Since the authors are not showing data during sunrise and sunset, it does not matter much.
* * *
We thank the reviewer for this insightful comment. The model results shown in Figure 4 were calculated assuming a plane-parallel atmosphere. We have conducted another calculation using the spherical geometry, where the sunrise and sunset times are dependent on altitude. We compare the resultant diurnal cycles with the one in the original manuscript below:

[Figure]

The above figure shows that the difference of the two diurnal cycles is much smaller than the spread of the data. Therefore, our conclusions remain unchanged when the spherical geometry is used.

In response to this comment, we added in the above figure in the new Appendix B.

**Box 1.5**

The linear fit in Figure 6 does not mean much, other than as a baseline, as there are two linear regimes, one from 07:00 to 13:00 and from 13:00 to 16:00 hours. Is there an explanation for the two regimes?

The linear fit in Figure 6 is used to compare with an independent measurement over Germany that is discussed in Section 3.3 [Fig. 3a of Sussmann et al., Atmos. Chem. Phys., 5, 2657–2677, 2005, https://doi.org/10.5194/acp-5-2657-2005]. Sussmann et al. reported a linear increasing rate of daytime $NO_2$ for the first time. To consistently compare with their analysis, we follow their definition of a linear fit through the daytime data. Our comparison with their value corroborates the findings over two different sites in mid-latitudes.

Our simulated total column $NO_2$ also shows a regime change before and after noon, although the simulated regime change is not as strong as in the observation. Thus, based on our photochemical model, the two linear regimes are likely due to the conversion of the reservoir species $N_2O_5$. Apart from the continuous production of NO through the reaction between $N_2O$ and $O(^1D)$, the photolysis rate of $N_2O_5$ peaks at local noon, causing a quadratic time dependence during the daytime and hence an apparent change in the linear regime before and after local noon. Indeed, in the original manuscript, we pointed out the important role of the $N_2O_5$ conversion in Line 232:

> "Figure 5 shows that the conversion between the reservoir and $NO_2$ dominates between 18 km and 34 km, consistent with the $NO_2$ diurnal cycles. Therefore, the secular $NO_2$ changes during daytime and nighttime are dominated by $N_2O_5$ conversions."

In response to this comment, we add the following in Line 308 of the revised manuscript:

> "Figure 5 shows that the conversion between the reservoir and $NO_2$ dominates between 18 and 34 km, consistent with the $NO_2$ diurnal cycles. In particular, the quadratic decreasing trend of the daytime $N_2O_5$ is consistent with the quadratic increasing trend of the daytime $NO_2$. Therefore, the secular $NO_2$ changes during daytime and nighttime are dominated by $N_2O_5$ conversions."

**Box 1.6**

On the instrument: What is the stray light?

The stray light is typically of the order of $10^{-4}$–$10^{-3}$, which is normally not high enough to affect the retrievals. In response to this comment, we added in Line 76:

> "The stray light is typically of the order of $10^{-4}$–$10^{-3}$."

Box 1.7

What is the SNR of each measurement?

We have estimated the SNR at full moon transit to be ~2900 and the SNR at solar transit to be ~4900. The SNR is estimated by taking the standard deviation of the difference of two consecutive spectra as the noise and the signal being the average intensity. During the low Sun/Moon observations the SNR is more difficult to measure directly. However, the fitting residuals are consistent with these estimates.

In response to this statement, we have added at the end of Section 2.1 (Line 94):

"We estimate the signal-to-noise ratio (SNR) by assuming that the standard deviation of the difference of two consecutive spectra is close to the noise and that the average intensity of the two consecutive spectra is the signal. As a result, the SNR at full moon and solar transits are ~2900 and ~4900, respectively. During the low sun/moon observations the SNR is more difficult to measure directly. However, the fitting residuals are consistent with these estimates."

Box 1.8

I know you are working with the standard QDOAS software, but could you give examples of the DOAS fitting and residuals.

We thank the reviewer for this suggestion. We have added a new Figure 2 in the revised manuscript.

[Figure]

The measured spectrum is shown by the black curve on the left panel. The fitted spectrum (red) is overlaid, and the residual spectrum (blue) is shown at the bottom. Four species are considered in the spectral fit: $NO_2$, $O_3$, $O_4$, and $H_2O$. The spectral fits are performed simultaneously in QDOAS. The red lines on the right column are the fitted spectra of the corresponding species. To visualize the signal-to-noise ratios, we add the residual spectrum (blue on the left panel) to individual fitted spectra, which are shown as the black spectra in the subpanels on the right.

Box 1.9

The writing is clear with no significant errors. The figures are clear and easy to understand. The title should be revised. The paper should be published with only minor changes as described above.

Once again, we thank the reviewer for his/her favour in our manuscript.

**Responses to Reviewer #2's comments**

Box 2.0

King-Fai Li et al 2020 present direct sun and direct moon NO2 measurements over a high altitude mostly unpolluted site, JPL-TMF near Wrightwood, CA, during 6 days (around full moon) in October 2018. They proposed to combine two Langley-like techniques to estimate amount of NO2 atmospheric absorption during the reference spectrum measurement time. The proposed approach takes advantage of 1-D photochemical model to estimate diurnal variation in NO2 and minimum Langley extrapolation technique to reduce the effect of NO2 pollution. Modeling results are compared with the measurements. Chemical reactions for different processes are shown. Accurate measurements of diurnal NO2 variation are important and the topic fits into the scope of the "Atmospheric Measurement Techniques" journal.

We thank Reviewer #2 for his/her time and useful comments that have greatly improved our manuscript.

Box 2.1

Not sufficient measurements (only 6 days) were presented in the paper to apply minimum Langley extrapolation technique (MLE). MLE is a statistical method and requires sufficient data to "accouter" conditions with "constant" vertical columns at each solar zenith angle. This threshold was not met during this study. MLE uses as low percentile for fitting as possible (within SNR) to capture background NO$_2$. Increasing percentile used for Langley fitting does not simply improves the statistics, as stated in the paper, but can significantly alter the result. This can be easily remediated by including more measurements, especially at this mostly clean site.

We thank the reviewer for this insightful comment. Since multiple instruments share the same dome at the TMF, the period we could run our instrument has been limited. Our preliminary measurements showed that 2 days away from the full moon would decrease the measured lunar intensity by ~20%, so we performed the measurement only when the moon was almost full to ensure a good signal-to-noise ratio (SNR). In addition to the full moon, we needed a non-cloudy atmosphere. The 6 consecutive days in October 2018 reported in this manuscript were the best period that satisfied these two conditions. It is our best interest to make more 24-hour measurements in the future.

  While we agree with the reviewer that a low percentile gives a better representation of the background NO$_2$ in a polluted site, the measurement over the TMF, as a clean site, may allow a higher percentile for an adequate estimation of the background on the TMF. As our back-trajectory analysis suggests, only one day out of six showed a sign of the urban source. Most vertical spread of the Langley plot is likely due to natural daily variability in the background. We therefore argue that the 20-percentile we used in the original manuscript still lies within the background variation. Nonetheless, we follow the reviewer's suggestion and have performed another Langley extrapolation using the 5-percentile baseline. The resulting equation is

$$y = (0.88 \pm 0.08)\, m\, x_a - (6.09 \pm 0.65) \times 10^{15} \quad \text{(5-percentile)}.$$

The y-intercept is now $0.32 \times 10^{15}$ more negative than the previous value obtained using the 20-th percentile in the original manuscript (which was $5.77 \pm 0.87 \times 10^{15}$) but they are well within the 2-$\sigma$ error. The new result is presented in Figure 3 of the revised manuscript:

[Figure]

Box 2.2

While the idea of improving estimation of slant column density in the reference spectrum using a 1-D photochemical model is appealing, the authors have not demonstrated that it provides a better result than MLE itself. Looking at the data in Fig. 2 and 3, MLE will most likely result in lower amount in the reference spectrum, and the final vertical columns will agree significantly better with the model diurnal change than the retrieved columns (but will have an offset). Authors need to show that the results are better than the MLE by itself, and for that more measurements are required. Note, that to determine amount in the reference spectrum, full moon is not needed, since the analysis is done on the direct sun data. It is not clear from the presented results if the error in the model simulations actually is smaller than the uncertainty in MLE. This needs to be demonstrated.

We thank the reviewer for this insightful comment. The simulated diurnal cycle is used to remove the asymmetry about noon in the total column $NO_2$. This asymmetry is natural, and it exists even under clean conditions and cannot be dealt with by the standard MLE (as explained in Section 2.3 of the original manuscript). To further illustrate the necessity of the removal of the diurnal asymmetry, we plot the observed total column $NO_2$ on a single day (e.g. Oct 25, 2018) against the air mass factor (AMF = $\sec\theta$) as in a standard MLE.

[Figure]

Panel (a) plots the observations on October 25, 2018 against the air mass factor (AMF = sec θ) as in a standard MLE. Based on our back-trajectory analysis, the atmosphere above TMF on October 25, 2018 should have little urban NO₂ contamination. Both solar (pale orange dots) and lunar (pale blue dots) data exhibit a U shape that is due to the secular increase and decrease during the daytime and the nighttime, respectively. For the solar data, the AM data lies on the lower arm of the U shape and the PM data lies on the upper arm. For the lunar data, the reverse is true: data before sunrise lie on the upper arm of the U shape and data after sunset lie on the lower arm.

To perform a Langley extrapolation for the data shown in Panel (a), one needs to decide which of the four arms to be used for the linear regression model $y = a\,\mathrm{AMF} + b$. The Principle of Minimum-amount suggests that we should start with the lowest arm, i.e. the daytime AM data. Note that in order to obtain the straight line passing through the 2-percentile baseline, we have ignored the points before noon (around 10 AM to 11:30 AM), i.e. points located around the bottom of the U-shape. If we use the observations between 6 AM and 10 AM, we obtain the purple line in Panel (a), which gives a $y$-intercept of $(-4.12 \pm 0.14) \times 10^{15}$ molecules cm⁻².

The above Langley extrapolation, however, does not take any of the daytime PM and all lunar data into account. In particular, the daytime PM data should also be used to define a minimum-amount profile, given the fact that the atmosphere was mostly clean on that day. Suppose we perform another Langley extrapolation using the daytime PM data between 12 PM and 5 PM (rose line). The resultant $y$-intercept is $(-5.25 \pm 0.27) \times 10^{15}$ molecules cm$^{-2}$ (2-$\sigma$), which is statistically different from the value obtained using the daytime AM data. A reasonable estimate of the $y$-intercept is then the average of the two values, which is $(-4.69 \pm 0.21) \times 10^{15}$ molecules cm$^{-2}$.

Finally, since the wind on the TMF is mostly downhill during autumn, the lunar data also correspond to a clean atmosphere and should also be used to derive the $y$-intercept. If we use all four arms in Panel (a), then the average value of the $y$-intercept is $(-4.36 \pm 0.25) \times 10^{15}$ molecules cm$^{-2}$, where the uncertainty is the root-mean-squares of the uncertainties of the four values.

In the above calculation, the ignorance of the data points near the bottom of the "U"-shape has excluded a large number of observations near local solar/lunar noon and thus the resultant $y$-intercept is biased by high zenith angles. It is not clear how the data near the solar/lunar noon may be kept in the standard MLE due to the assumption of the linearity in AMF. As a result, a zenith angle-dependent Langley extrapolation model needs to be developed.

The above example shows that the determination of the $y$-intercept of the standard MLE is not straightforward when (i) the background NO$_2$ has secular trends in daytime and nighttime and (ii) the daytime and nighttime abundances are different before and after the terminator. In contrast, the modified MLE (MMLE) approach we have developed in this work minimizes the background diurnal asymmetry, so that the "regularized" data points almost form a straight line (Panel b) when they are plotted against the modelled diurnal cycle. The linear regression model $y = a\,m\,x_a + b$, where $m\,x_a$ is the modelled slant column NO$_2$, can be applied to all data points, regardless of the time of the day or whether the data point is a solar or lunar measurement. With this MMLE, the regressed $y$-intercept is $(-5.22 \pm 0.14) \times 10^{15}$ molecules cm$^{-2}$, which is statistically different from the average of the values derived from the four arms in the standard MLE approach.

The issue with the standard MLE is exacerbated when observations on multiple days are plotted against the AMF. The U-shape may be smeared vertically into a continuum (Panel c). The smearing, in our case, are primarily due to natural variability of the background, except for October 27 when total column NO$_2$ appears above the continuum of the daytime data due to the urban pollution. As a result, while we are still able to define the minimum-amount profile for the daytime AM data, the determination of the minimum-amount profiles of the daytime PM and the lunar data are difficult. This leaves us the daytime AM data alone for the Langley extrapolation (red line) but, as shown above, the resultant $y$-intercept [$(-4.10 \pm 0.46) \times 10^{15}$ molecules cm$^{-2}$] may be biased.

In contrast, the observed data points still almost form a straight line in the MMLE approach when they are plotted against the modelled diurnal cycle (Panel d). This allows the determination of the minimum-amount profile using all solar and lunar measurements (raspberry line). The resultant $y$-intercept, $(-6.09 \pm 0.65) \times 10^{15}$ molecules cm$^{-2}$, is again statistically different from the one obtained using the standard MLE approach.

As pointed out in Box 2.1 and Box 2.8, 2 days away from the full moon would decrease the lunar signal by 20%. We thus focused on full moon to ensure a high SNR for testing our instrument.

In response to this comment, we have put the above argument in a new Appendix A of the revised manuscript.

Box 2.3

It is unclear what benefits this approach (1D stratospheric model) will have under the "persistent" pollution levels when the total $NO_2$ abundance is dominated by anthropogenic emissions at all times.

As explained in Box 2.3 (and Appendix A of the revised manuscript), the standard MLE may have a bias in the y-intercept because the total column $NO_2$ has a natural diurnal asymmetry in the background, regardless of the pollution level. The removal of the natural diurnal asymmetry using the 1-D stratospheric model is thus necessary for $NO_2$. The removal is also necessary to monitor the pollution level.

In addition, Table Mountain has been providing ground-based measurements for validating satellite retrievals because the site is away from the Los Angeles area and the tropospheric contribution to $NO_2$ is generally low, even during summer daytime when upslope wind from the southwest (through LA downtown) is the strongest compared to other seasons, as discussed in one of our previous publications (Wang et al., JGR 2010, 10.1029/2009JD013503, cited in the original manuscript). We thus anticipate that pollution does not pose much of a problem in our MMLE. Furthermore, the more data we have in the future, the more accurate will be the minimum-amount baseline of the Langley plot, which can be directly compared with the 1-D stratospheric model. Indeed, in the original manuscript, we wrote (Line 136):

> "When a large number of measured $NO_2$ columns on clean and polluted days are plotted together against $m\, x_a(m)$, the baseline of the scattered data may be considered as the background $NO_2$ diurnal cycle in a clean atmosphere (Herman et al., 2009)."

In response to this comment, we clarify in the revised manuscript (Line 190):

> "Our measurements made during October (a non-summer season) were mostly under unpolluted conditions. Thus, we applied the MMLE to derive a baseline for an estimation of the background $NO_2$ diurnal cycle, which is then used in the regression with the modelled diurnal cycle."

Box 2.4

Description of the DOAS fitting settings is not sufficient. What are polynomial orders (e.g. broad band, offset, wavelength shift), what sources of other gases cross sections and at what temperatures were used in the analysis? Were $NO_2$ cross sections at all five temperatures used in the retrieval? If yes, how they were fitted and combined? Why this fitting window was selected (430 and 468 nm)? How exactly air mass factor was calculated? What is the DOAS fitting quality of $NO_2$ from direct sun and direct moon (residual OD)?

We use 3rd order polynomials for broadband and offset. The $NO_2$ cross section is from Nizkorodov et al. (2004) for 215 K, 229 K, 249 K, 273 K, and 299 K, as discussed in the original manuscript. The $O_3$ cross section is from Serdyuchenko et al. (2014) for 11 temperature references ranging from 193 K to 293 K. The $O_4$ cross section is from Thalman and Volkamer 2013 at 273 K, and

H$_2$O cross section at 296 K is from HITRAN 2016. All five cross sections were used to create a single NO$_2$ reference. The yearly average from the TMF temperature LIDAR are used to derive a reference for each altitude level by linear interpolation between each adjacent cross-section, which is also adjusted for pressure broadening using the results of Nizkorodov et al. (2004). Each level's reference is then multiplied by a weight which is proportional to the standard atmosphere and then summed to obtain a single reference used in the fitting. A similar procedure was used for the O$_3$ reference; for O$_4$ and H$_2$O only a single reference was used.

As shown in Figure 3 of Spinei et al. (2014), the 430–468 nm window has stronger NO$_2$ absorptions relative to other wavelengths in the 411–475 nm range. In addition, this window also has less interfering absorption from other species. These two factors add up to increase the accuracy of the DOAS spectral fit.

The air mass is calculated using secant of the solar/lunar zenith angle, sec(VZA). Herman et al. (2009) considered an altitude correction in this equation. However, the correction is generally negligible except for VZA > 80° but we do not make measurements at those VZAs.

The new Figure 2 in the revised manuscript shows an example of the spectral fit.

In response to this comment, we have added the above descriptions in the revised Section 2.2:

"The differential slant column NO$_2$ is retrieved by fitting the ratioed spectrum in a smaller window between 430 and 468 nm. This window has stronger NO$_2$ absorptions relative to other wavelengths in the instrument range (411–475 nm); see Figure 3 of Spinei et al. (2014). In addition, this window also has less interfering absorption from species other than the O$_3$, O$_4$ (O$_2$ dimer), and H$_2$O (see below).

The spectral fitting is accomplished through the Marquardt-Levenberg minimization using QDOAS retrieval software (http://uv-vis.aeronomie.be/software/QDOAS/). The highly spectrally resolved NO$_2$ absorption cross sections at $T = 215$ K, $229$ K, $249$ K, $273$ K, $298$ K, and $299$ K based on Nizkorodov et al. (2004) are convolved to the instrument resolution using the instrument line shape function and the Voigt line shape prior to its use in QDOAS. The yearly average from the TMF temperature LIDAR measurements are used to derive a reference for each altitude level by linear interpolation between each adjacent cross-section, which is also adjusted for pressure broadening using the results of Nizkorodov et al. (2004). We use 3rd order polynomials for broadband and offset. All five cross sections were used to create a single NO$_2$ reference. Each level's reference is then multiplied by a weight which is proportional to the standard atmosphere and then summed to obtain a single reference used in the fitting. In addition to NO$_2$, other absorptions by O$_3$, O$_4$ (O$_2$ dimer), and H$_2$O in the same spectral window are simultaneously retrieved. The O$_3$ cross section is from Serdyuchenko et al. (2014) for 11 temperature references ranging from 193 K to 293 K. Like NO$_2$, all 11 cross-sections are used in the spectral fitting for O$_3$. In contrast, for O$_4$ and H$_2$O, only a single temperature reference is used. The O$_4$ cross section is from Thalman and Volkamer (2013) at 273 K. The H$_2$O cross section at 296 K is from HITRAN 2016 (Gordon, 2017). Figure 2 shows an example of a fitted spectrum on October 24, 2018. The NO$_2$ abundance retrieved from QDOAS is the desired differential slant column NO$_2$ relative to our chosen reference spectrum.

The air mass factor is calculated using secant of the solar/lunar zenith angle. Herman et al. (2009) considered an altitude correction of the air mass factor. The altitude

correction is generally negligible except for zenith angles ≥ 80° but we do not make measurements at those zenith angles (see Section 2.1)."
* * *
Box 2.5

No error budget is presented for the measured NO₂ columns.

We considered two major sources of error: the fitting residual of the DOAS spectral fit and the error of the y-intercept of the Langley extrapolation. The errors from the QDOAS fitting residual generally lies between than $0.1\times10^{15}$ cm$^{-2}$ and $0.6\times10^{15}$ cm$^{-2}$ (2-σ) for all zenith angles, which is now shown in an inset of Figure 3 of the revised manuscript:

[Figure]

The error of the y-intercept of the Langley extrapolation is $\pm0.65\times10^{15}$ cm$^{-2}$ (2-σ), which is also added in the revised Figure 3 (see the figure above). The root-mean-square of these two sources of error gives an estimate of a total error of $\sim0.9\times10^{15}$ cm$^{-2}$ (2-σ).

In response to this comment, we added in the revised manuscript in Line 143:

> The 2-$\sigma$ uncertainty due to the spectral fitting residual lies between $0.1\times10^{15}$ molecules cm$^{-2}$ and $0.6\times10^{15}$ molecules cm$^{-2}$, with a mean of $\sim0.4\times10^{15}$ molecules cm$^{-2}$. The distribution of the retrieval uncertainty is shown in Figure 2 (inset).

and in Line 198:

> "We estimate the total retrieval uncertainty to be the root-mean-square of the spectral fitting uncertainty and the uncertainty in $y_0$, which is $0.8\times10^{15}$ molecules cm$^{-2}$ (2-$\sigma$)."

┌─────────────────────────────────────────────────────────────┐
│                         Box 2.6                             │
│  The paper in general reads more like a modeling paper then the measurement paper.  │
└─────────────────────────────────────────────────────────────┘

Thanks for this comment. Actually, Reviewer #1 has an opposite comment. The aim of this paper is to present a new instrument for measuring daytime and nighttime $NO_2$ column. Perhaps the confusion may be due to the frequent appearances of models in the retrieval strategy and the back-trajectory calculations. However, the 1-D stratospheric model is mainly used to assist, not determine, the retrieval (by minimizing the diurnal asymmetry). The retrieval per se is still observation-based, in contrast to the common Bayesian-based approach where the statistics of the *a priori* model is also used to constrain the retrieved value.

In response to this comment, in the revised manuscript, we have removed the phrase "model-based" from the name of our method. Instead, our method is now called "the modified minimum-amount Langley extrapolation" or MMLE in short.

┌─────────────────────────────────────────────────────────────┐
│                         Box 2.7                             │
│  There are routine $NO_2$ stratospheric measurements conducted by the NDACC stations (zenith  │
│  sky DOAS) and total column measurements of $NO_2$ using direct sun and direct moon within    │
│  Pandonia Global Network. They should be mentioned in the review of $NO_2$ measurements. In   │
│  general, citations tend to include mostly early works and not give current status.           │
└─────────────────────────────────────────────────────────────┘

We thank the reviewer for mentioning these references. We are aware of them. Indeed, we have used NDACC $NO_2$ data in our recent publication (Wang et al., Solar 11-Year Cycle Signal in Stratospheric Nitrogen Dioxide—Similarities and Discrepancies Between Model and NDACC Observations, Solar Phys., doi:10.1007/s11207-020-01685-1, 2020, cited in the revised manuscript). In addition, MF-DOAS and Pandora were used in our earlier publication (Wang et al., 2010, cited in the original manuscript.).

In response to this comment, we have added a number of references involved in NDACC and PGN. In particular, the statement in Line 24:

"$NO_2$ column abundance has been measured using ground-based instruments since the mid-1970s [Network for the Detection of Atmospheric Composition Change (NDACC), http://www.ndacc.org] …"

has been revised as

"$NO_2$ column abundance has been measured using ground-based instruments since the mid-1970s [Network for the Detection of Atmospheric Composition Change (NDACC), http://www.ndacc.org] (e.g., Hofmann et al., 1995; Piters et al., 2012; Roscoe et al., 1999; Roscoe et al., 2010; Vandaele et al., 2005; Kreher et al., 2020) …"

In addition, the statement in Line 59:

"Other techniques, such as balloon-based *in situ* measurements (May and Webster, 1990; Moreau et al., 2005), balloon-based solar occultations (Camy-Peyret, 1995) and groundbased multi-axis DOAS (MAX-DOAS; Hönninger et al., 2004; Sanders et al., 1993) have also been employed to further characterize the vertical distributions of NO2."

has been revised as

"Other techniques, such as balloon-based *in situ* measurements (May and Webster, 1990; Moreau et al., 2005), balloon-based solar occultations (Camy-Peyret, 1995), as well as ground-based multi-axis DOAS (MAX-DOAS; Hönninger et al., 2004; Sanders et al., 1993), multi-functional DOAS, and Pandora (Herman et al., 2009; Spinei et al., 2014) that have been actively involved in NDACC and the Pandonia Global Network (Kreher et al., 2020), have also been employed to further characterize the vertical distributions of NO2."

Box 2.8
It is unclear how the lunar measurement where taken. Lunar irradiance is about $10^6$ lower than solar irradiance. In this study, integration time for sun measurements is 16 times shorter than for moon. Difference in lunar vs solar measurements (diffusers, filters, etc), and what effect it has on spectrometer illumination should be presented. Target signal-to-noise ratio stated.

We apologize for the confusion. For the sunlight measurement, we insert a diffuser plate to reduce the solar throughput by a factor of $\sim 1.3 \times 10^{-5}$ and protect the instrument. The diffuser plate is not used during the moonlight measurement. Since the sun is ~400,000 times the intensity of the full moon, the ratio between the light hitting our detector for solar noon (after inserting the diffuser and the filter) and lunar noon during the full moon is ~5. Thus, in order to maintain an approximately constant solar and lunar signal-to-noise ratio and fitting residuals, we need to vary slightly the exposure time during specific times of solar and lunar noon, typically around ~3 s for lunar noon and ~0.6 s for solar noon, giving a ratio of ~5 to balance out the photon counts mentioned above. In the original manuscript, we wrote the statement (Line 77)

"The exposure time was 4 s and 0.25 s during the lunar/solar noon observations, respectively."

The 4 s and 0.25 s are the full range of exposure times between which we varied during that week of measurement in October 2018, but the writing of this statement may be confusing. The exposure times were not constant during the measurement.
In response to this comment, the above quoted statement has been revised to (Line 86)

"When direct moonlight is measured, the diffuser plates are removed. Since the sun is ~400,000 times the intensity of the full moon, the ratio between the light hitting our detector for solar noon (after inserting the diffuser plates) and lunar noon during the full moon is ~5. To maintain an approximately constant solar and lunar signal-to-noise ratio and fitting residuals, we vary the exposure time during specific times of solar and lunar noon, typically around ~3 s for lunar noon and ~0.6 s for solar noon, giving a ratio of ~5 to homogenize the solar and lunar photon counts mentioned above."

---

## Editor Comment (EC1) · Ralf Sussmann (Editor) · 8 Dec 2020

I do not agree with your arguments in reply to ____ Box 1.5 The linear fit in Figure 6 does not mean much, other than as a baseline, as there are two linear regimes, one from 07:00 to 13:00 and from 13:00 to 16:00 hours. Is there an explanation for the two regimes? ______

- Fig. 3a in Sussmann et al. 2005 does not show indications for any measurable non-linear diurnal change

- Your modeled diuarnal variation is much closer to a linear behavior than your measurement results.

This means, obviously the main reason for your observed non-linear behavior is a different (probably experimental) one, i.e., your trying to explain via modeling is flawed or can at best be used as a minor, partial explanation. So you should throughly discuss possible reasons which could make the measurements erroneously look non-linear, i.e., please quantitatively estimate possible measurement articfacts like airmass/zenith angle dependencies.

---

## Author Response (AR2)

**TABLE OF CONTENTS**

**Comments from Reviewer #1**

Box 1.1
Review #1
Anonymous during peer-review:                                     Yes
Anonymous in acknowledgements of published article:      Yes

Recommendation to the editor
1) Scientific significance
Does the manuscript represent a substantial contribution to scientific progress within the scope
of this journal (substantial new concepts, ideas, methods, or data)?                    Good

2) Scientific quality
Are the scientific approaches and applied methods valid? Are the results discussed in an
appropriate and balanced way (consideration of related work, including appropriate
references)? Note that papers do not necessarily need to be long to be scientifically sound.
          Fair

3) Presentation quality
Are the scientific results and conclusions presented in a clear, concise, and well structured way
(number and quality of figures/tables, appropriate use of English language)?        Good

For final publication, the manuscript should be
accepted subject to minor revisions

Suggestions for revision or reasons for rejection (will be published if the paper is accepted for
final publication)

We thank the reviewer for providing encouraging comments.

Box 1.2
I agree with the comments of the other reviewer that this seems more like a modeling paper than a
measurements paper with only a little description of the instrument characteristics and a very
small number of measurements. Error estimates are missing.

We thank the reviewer for this very constructive comment. We would like to point out that this is the first day/night contiguous measurement of $NO_2$ column being published, which provides an important test of photochemical models. Our pilot study with a week of data successfully demonstrates the feasibility of retrieving both daytime and nighttime $NO_2$ abundances using the grating spectrometer measurements and the applicability of the modified Langley method, as well as the qualitative agreement with the photochemical model. In the last review cycle, we added a new Figure 2 showing the spectral fit and an inset of the new Figure 3 that shows the errors of the QDOAS fitting. We also added the signal-to-noise ratios of the measurements and added the uncertainty of the parameters of the Langley extrapolation. The new information has been

presented in the revised manuscript but was not present in the original discussion paper published on the AMTD website. We hope that the new information provided in the last review cycle has addressed this comment. We are happy to provide more information if the reviewer would like to see additional error estimates.

> **Box 1.3**
>
> Even with intrusion from nearby cities, the $NO_2$ amounts are very low on the "worst" day, October 27, about 0.2 DU, barely above stratospheric values. This paper is totally about the stratospheric behavior of $NO_2$ and should be described as such.

We greatly appreciate this comment. We have replaced "total column $NO_2$" by "stratospheric column $NO_2$" in most places (including the title), except when we discussed the measurement on October 27, 2018 when the tropospheric contribution was significant, we simply call it "$NO_2$". We hope that these changes would help clarify our measurements.

> **Box 1.4**
>
> Even with intrusion from nearby cities, the NO2 amounts are very low on the "worst" day, October 27, about 0.2 DU, barely above stratospheric values. This paper is totally about the stratospheric behavior of NO2 and should be described as such. The 1D modeling is adequate for stratospheric behavior and is useful as an indicator that the measurements are reasonable. The non-linear behavior compared to the model during the daytime is not explained either as a chemistry result or as an instrument problem. Aside from October 27, the curvature is repeatable on multiple measurements (Figs. 3 and 6) and should be explained if it is possibly an instrument problem.

We believe that the reviewer is referring to Figures 3 and 6 of the original discussion paper published on the AMTD website. In the revised manuscript, the corresponding figures are Figures 4 and 7.

We greatly appreciate this comment. Since the Editor has the same concern, we reinvestigated our Langley extrapolation. We found that the curvature of the daytime data was likely due to two artifacts (please see Box E.2 on page 7 of this response letter for a more detailed discussion): (1) a low percentile that defines the baseline of the Langley plot and (2) the bias of the extrapolation due to the sparse data points at high air mass factors. After correcting these artifacts, the curvature is significantly reduced. We thus believe that the curvature is unlikely due to an instrument problem. We have modified our Figures 3, 4, 7, A1 and B1 accordingly.

> **Box 1.5**
>
> The new figures and some of the discussion given in the reply to reviewers must be part of this paper before publication.

We believe that the reviewer is referring to the new figures we showed in our previous response letter. Those figures were included in the last revised manuscript. The original discussion paper published on the AMTD website does not have those new figures. We are happy to provide direct access to the revised manuscripts if necessary.

Box 2.1
Review #2
Anonymous during peer-review:                                    Yes
Anonymous in acknowledgements of published article:     Yes
Recommendation to the editor
1) Scientific significance
Does the manuscript represent a substantial contribution to scientific progress within the scope
of this journal (substantial new concepts, ideas, methods, or data)?                    Good

2) Scientific quality
Are the scientific approaches and applied methods valid? Are the results discussed in an
appropriate and balanced way (consideration of related work, including appropriate
references)? Note that papers do not necessarily need to be long to be scientifically sound.
        Good

3) Presentation quality
Are the scientific results and conclusions presented in a clear, concise, and well structured way
(number and quality of figures/tables, appropriate use of English language)?        Good

For final publication, the manuscript should be
accepted subject to technical corrections

Suggestions for revision or reasons for rejection (will be published if the paper is accepted for
final publication)

We greatly appreciate the reviewer for the positive evaluation of our manuscript.

Box 2.2
There are a few minor things that need some clarification:

- 3rd order polynomial used for offset correction in DOAS fitting is higher than typical (1st
order). The offset is mainly used to correct spectra for instrumental stray light and 3rd order
seams rather high for this wavelength range.

We thank for this thoughtful comment. We actually tested our algorithm with a linear baseline. We found that the 3rd order polynomial gave a smaller residual than the linear baseline while it does not overfit the narrow $NO_2$ absorption features in our spectral window. Regarding the order of the polynomial, some studies, such as Herman et al. (2009) who also retrieved $NO_2$ column from DOAS measurements, use 4th or higher order polynomials for wider spectral windows. Our choice of the 3rd order polynomial is a good compromise between residual reduction and overfitting.

In response to this comment, we have added the following statement in Line 121:

"Some studies, such as Herman et al. (2009), use 4th or higher order polynomials for wider spectral windows. Since the $NO_2$ absorption features are much narrower than our spectral window (430–468 nm), the broad shape of the 3rd order polynomial does not affect the $NO_2$ retrievals. In addition, for our spectral window, we tested our retrieval algorithm using a linear baseline and we concluded that a 3rd order polynomial reduces the residuals more effectively than a linear baseline."

Box 2.3

description how NO2 mol. absorption cross section at an effective temperature is created from the climatological temperature profiles and standard NO2 profile should be improved. It will be good to state what that final effective NO2 temperature is for that location.

Our $NO_2$ cross section reference assumes the yearly average temperature profile at TMF and a low level of free tropospheric $NO_2$, thus our effective temperature is 231 K. To test the sensitivity of these assumptions we considered two extreme cases. A cooler atmosphere with a lower partition of $NO_2$ in the free trop and a warmer atmosphere with a higher partition of $NO_2$ in the free troposphere. The effective temperatures of these two cases are estimated by 229 K and 249 K, respectively. The difference between retrievals using these extreme cases is ~5%; the regular variation of temperature and tropospheric $NO_2$ at TMF is well within estimates.

In response to this comment, we have added the following statement in Line 126:

"Our $NO_2$ cross section reference assumes the yearly average temperature profile at TMF and a low level of free tropospheric $NO_2$. The effective temperature of the $NO_2$ absorption cross section used in the work is 231 K. To test the sensitivity of these assumptions we considered two extreme cases: (i) a cooler atmosphere with a lower partition of $NO_2$ in the free troposphere and (ii) a warmer atmosphere with a higher partition of $NO_2$ in the free troposphere. The effective temperatures of these two cases are estimated by 229 K and 249 K, respectively. The difference between retrievals using these extreme cases is ~5%; the regular variation of temperature and tropospheric $NO_2$ at TMF is well within estimates."

Box 2.4

QDOAS version should be stated.

For this work we have used the QDOAS 3.2 (2017). We have modified the following statement in Line 115 from

"The spectral fitting is accomplished through the Marquardt-Levenberg minimization using QDOAS retrieval software (http://uv-vis.aeronomie.be/software/QDOAS/)"

to

"The spectral fitting is accomplished through the Marquardt-Levenberg minimization using QDOAS 3.2 (released in September 2017) retrieval software ([http://uv-vis.aeronomie.be/software/QDOAS/](http://uv-vis.aeronomie.be/software/QDOAS/))"
* * *
Box 2.4

Figure 2 shows an example of the DOAS fit, but it is does not indicate at what time and date the spectrum and the reference were taken. Is this solar or lunar data fit? It might be interesting to see an example of both, since the illumination of the instrument changes and might impact the quality of the fit.
* * *
The sample spectral fit shown in Figure 2 was a lunar measurement at 7:25 PM on October 24, 2018, corresponding to an air mass factor of 2.21. This information can now be found in the caption of Figure 2 of the latest revised manuscript. Solar measurements generally have better signal-to-noise ratios, so we believe that the lunar sample in Figure 2 is very representative of the quality of our spectral fit. For clarity of the paper, we intend to include only 1 figure of spectral fit. If there is a strong interest in a similar spectral fit of a solar measurement, we are more than happy to add another figure in the final version of the manuscript.
* * *
Box 2.5

I recommend rephrasing: Other techniques, such as balloon-based in situ measurements (May and Webster, 1990; Moreau et al., 2005), balloon-based solar occultations (Camy-Peyret, 1995), as well as ground-based Differential Optical Absorption Spectroscopy: MAX-DOAS (Hönninger et al., 2004; Sanders et al., 1993), Direct Sun DOAS (Herman et al., 2009; Spinei et al., 2014, more updated references here) that have been actively applied in NDACC and the Pandonia Global Network. The DOAS techniques have also been employed to further characterize the vertical distributions of NO2 (Kreher et al., 2020).
* * *
We greatly appreciate this suggestion. We have modified the statement in Line 50 accordingly.

Box E.1

Editor's comment

Dear authors,

there are 2 anonymous reveiws which you should carefully address in all points. In particular, I share the view that it is not a total column paper but your results are only relevant for the stratosphere and this should be addressed via a changed title.

We thank Reviewer #1 and this Editor for this suggestion. We have modified the title accordingly.

Box E.2

Furthermore, please carefully address these additional major concerns from my side:

- I do not agree with your arguments in reply to Box 1.5 The linear fit in Figure 6 does not mean much, other than as a baseline, as there are two linear regimes, one from 07:00 to 13:00 and from 13:00 to 16:00 hours. Is there an explanation for the two regimes?
- Fig. 3a in Sussmann et al. 2005 does not show indications for any measurable non-linear diurnal change
- Your modeled diurnal variation is much closer to a linear behavior than your measurement results. This means, obviously the main reason for your observed non-linear behavior is a different (probably experimental) one, i.e., your trying to explain via modeling is flawed or can at best be used as a minor, partial explanation. So you should thoroughly discuss possible reasons which could make the measurements erroneously look non-linear, i.e., please quantitatively estimate possible measurement artifacts like airmass/zenith angle dependencies.

We would like to thank the Editor for providing this critical comment. We agree with the Editor's observations about the difference between our data and his own data. We thank the Editor for his patience during the COVID-19 pandemics and his willingness to handle our revisions and improve our manuscript.

We spent a long time to investigate the potential factors in our spectral retrievals and the Langley extrapolation that may have caused the artifacts in the diurnal cycle. We found that two pre-processings of the Langley extrapolation were likely the causes.

(i) A low percentile for determining the baseline

In our previous manuscript, we used a 5-percentile to define the baseline of the data cluster in Figure 3, which was then used as a clean-atmosphere baseline for the modified Langley method. However, this 5-percentile is too low compared with the 10% uncertainty of the QDOAS spectral retrieval. Thus the 5-percentile appears to be inconsistent with the uncertainty of the data points in Figure 3 and leads to uncertainty in the Langley extrapolation.

For the above reason, we have instead used a 10-percentile to define the baseline in the latest revision.

(ii) The bias in the Langley extrapolation due to sparse data points at high air mass factors

The accuracy of the Langley extrapolation critically depends on the accuracy of the diurnal baseline obtained above. Statistically, at least more than 10 data points in a bin are required to determine the 10-percentile. While we made temporally dense measurements during daytime and nighttime (at intervals less than ~20 minutes, as discussed in §2.1), the application of the air mass factor $m = \sec\theta$ in both the $x$- and $y$-axes of Figure 3 significantly stretch the time intervals at high air mass factors. The number of data points in the bins thus drops progressively by a factor of ~2: the data counts drop exponentially from 431 in the first bin, $(4.5–6)\times10^{15}$ molecules cm$^{-2}$, to only 12 in the 8th bin, $(1.5–1.65)\times10^{16}$ molecules cm$^{-2}$. The determination of the 10-percentile for bins with centers greater than $1.5\times10^{16}$ molecules cm$^{-2}$ is then subject to large uncertainties. Since mathematically, the 10-percentiles at high air mass factors (i.e. at the edge of the data distribution) have higher effects on a linear fit, the resultant Langley extrapolation would be strongly biased by the uncertainties of the 10-percentiles at high mass factors. Thus, to obtain a linear fit for the Langley extrapolation, we apply more weights to bins with more data counts. This definition of the weights should mimic the reduction of the variance of a sample mean by the factor of $\frac{1}{N}$ (or $\frac{1}{\sqrt{N}}$ for the standard deviation of a sample mean). Therefore, we define the weight as unity for the first bin, $(4.5–6)\times10^{15}$ molecules cm$^{-2}$. The weight for the second bin is the ratio of the data counts of this bin over the first bin. The weight for the third bin is the ratio of the data counts of this bin over the second bin, etc. The weighted linear fit obtained using these weights is used for the Langley extrapolation. The new Figure 3 (shown below) compares the Langley extrapolations using the weighted (solid red line) and unweighted linear fit (dashed red line). Since the 10-percentiles at high air mass ($\geq 1.5\times10^{16}$ molecules cm$^{-2}$) are generally overestimated due to insufficient data counts, the unweighted linear fit tends to have a steeper slope, leading to a ~15% higher reference column ($5.44\times10^{15}$ molecules cm$^{-2}$) relative to that obtained using the weighted linear fit. This overestimation of the reference column created the artifact in the diurnal cycle due to the normalization factor $m^{-1}(y + y_0)$.

[Figure]

After correcting the above pre-processing, we obtained a reference column of $4.74 \times 10^{15}$ molecules $cm^{-2}$. We revised Figures 3, 4, 7, A1 and B1 using the corrected reference column. We specifically reproduce Figure 7 below, which estimates the linear diurnal increase rate:

[Figure]

The curvature is significantly reduced, and the diurnal cycle is much more linear. Our result shows that the shape of the $NO_2$ diurnal profile is very sensitive to the extrapolated intercept of the Langley plot, which has not been emphasized in the literature. We hope that the corrected results have addressed the concerns by the Editor and Reviewer 1.

In response to this comment, we have replaced the 5-percentiles used in previous Figures 3 and A1 by the 10-percentiles in the latest revision. We have added the following paragraph in Line 195 of the latest revision:

"Note that the data points are sparsely distributed at high air mass factors in Figure 3. This is because while the measurements were made at relatively uniform time intervals, the air mass factor $m = sec\theta$ effectively stretch the time intervals at high air mass factors. The number of data points in the bins drops progressively by a factor of ~2: the data counts drop exponentially from 431 in the first bin, $(4.5-6) \times 10^{15}$ molecules $cm^{-2}$, to only 12 in the bin $(1.5-1.65) \times 10^{16}$ molecules $cm^{-2}$. The determination of the 10-percentile for bins with centers greater than $1.5 \times 10^{16}$ molecules $cm^{-2}$ is then subject to large uncertainties. Since mathematically, the 10-percentiles at high air mass factors (i.e. at the edge of the data distribution) have higher effects on a linear fit, the resultant Langley extrapolation would be strongly biased by the uncertainties of the 10-percentiles at high mass factors. Thus, to obtain a linear fit for the Langley extrapolation, we apply more weights to bins with more data counts. This definition of the weights should mimic the reduction of the variance of a sample mean by the factor of $\frac{1}{N}$ (or $\frac{1}{\sqrt{N}}$ for the standard deviation of a sample mean).

Therefore, we define the weight as unity for the first bin, $(4.5–6)\times10^{15}$ molecules cm$^{-2}$. The weight for the second bin, $(6–7.5)\times10^{15}$ molecules cm$^{-2}$ is the ratio of the data counts of this bin over the first bin. The weight for the third bin is the ratio of the data counts of this bin over the second bin, etc. The weighted linear fit obtained using these weights is used for the Langley extrapolation. Figure 3 compares the Langley extrapolations using the weighted (solid red line) and unweighted linear fit (dashed red line). Since the 10-percentiles at high air mass ($\geq 1.5\times10^{16}$ molecules cm$^{-2}$) are generally overestimated due to insufficient data counts, the unweighted linear fit tends to have a steeper slope, leading to a ~15% higher reference column ($5.44\times10^{15}$ molecules cm$^{-2}$) relative to the weighted linear fit. This overestimation of the reference column may create an artifact in the diurnal cycle due to the normalization factor $m^{-1}(y + y_0)$."